# Towards development of a high-strength stainless Mg alloy with Al-assisted growth of passive film

Qingchun Zhu[1], Yangxin Li [1]✉, Fuyong Cao[2], Dong Qiu [3], Yao Yang[1], Jingya Wang[1], Huan Zhang[1], Tao Ying[1], Wenjiang Ding[1] & Xiaoqin Zeng [1]✉

Magnesium alloys with high strength and excellent corrosion resistance are always sought-after in light-weighting structural components for automotive and aerospace applications. However, for most magnesium alloys that have a high specific strength, they usually have an inferior corrosion resistance and vice versa. In this work, we successfully develop a Mg-11Y-1Al (wt. %) alloy through conventional casting, solution treatment followed by extrusion. The overall properties of this alloy feature with a corrosion rate lower than $0.2 \, \text{mm} \, \text{y}^{-1}$, high yield strength of 350 MPa and moderate tensile elongation of 8%, the combination of which shows competitive advantage over other comparative magnesium alloys in the literature. It is found that a thin and dense protective film of $Y_2O_3$/$Y(OH)_3$ can be fast developed with the aid of $Al_2O_3$/$Al(OH)_3$ deposition to isolate this alloy from further attack of corrosion medium. Meanwhile, the refined grains, weak texture and activation of non-basal slip systems co-contribute to the high strength and good ductility. Our findings are expected to inspire the design of next-generation high performance magnesium alloys.

As the lightest metallic structural materials, magnesium (Mg) alloys have great potential in industry where lightweighting is the key avenue to reduce the energy consumption and carbon footprint, but their inferior mechanical properties at room temperature and weak resistance against corrosion are the two major bottlenecks limiting their applications[1-5]. It is well acknowledged that the low strength and limited ductility of Mg originate from its intrinsic weak binding force and insufficient slip systems at room temperature while the poor corrosion resistance is mainly attributed to its low corrosion potential and the porous corrosion product on its surface[3,6-8]. In order to combat these two long-standing problems of Mg alloys, a variety of strategies have been developed spanning from alloy design to manufacturing process in the past decades.

Whilst the mechanical properties of Mg alloys can be effectively improved through micro-alloying, in particular adding rare earth

(RE) elements and/or grain refinement[9-11], improving their corrosion resistance is a different story. The corrosion rate of high-purity magnesium (HP-Mg, Mg ≥ 99.99 wt. %) is $0.3$-$0.5 \, \text{mm} \, \text{y}^{-1}$ [12], which is reasonably good, but its application is very limited due to its very low yield strength (<30 MPa)[13]. When alloying elements are being used to improve the mechanical properties, the corrosion resistance is usually compromised[14]. Only a few reports found that the corrosion rate of Mg alloy is lower than that of HP-Mg[4,15,16]. When a small amount of Nd[17] or Y/Ca[18] was added into AZ91-series alloy, the corrosion rate can be alleviated through reducing the potential difference between the secondary phases and the magnesium matrix. Another recent study[19] indicated that trace addition of Ca into ultra-pure magnesium could slow down the corrosion rate by controlling impurities and inducing a protective film. Whilst the corrosion resistance of the abovementioned alloys[17-19] is marginally

[1]National Engineering Research Center of Light Alloy Net Forming and State Key Laboratory of Metal Matrix Composite, Shanghai Jiao Tong University, Shanghai, PR China. [2]Center for Marine Materials Corrosion and Protection College of Materials, Xiamen University, Xiamen, PR China. [3]Centre for Additive Manufacturing, School of Engineering, RMIT University, Melbourne, VIC, Australia. ✉e-mail: astatium@sjtu.edu.cn; xqzeng@sjtu.edu.cn

better than HP-Mg, their mechanical properties are far from satisfaction.

The service environment of Mg alloys usually requires high standards of both mechanical and corrosion resistance properties to ensure good structural integrity and durability. Unfortunately, these two requirements often contradict to each other. For example, the corrosion rate of Mg-RE binary alloys increases exponentially with increasing RE content[20], which is attributed to the formation of galvanic cells between RE-containing secondary phases and the magnesium matrix[20–22]. Although some researchers have proposed strategies to simultaneously improve the strength and corrosion resistance with alleviating the side-effect of secondary phases[23–25], and a promising strength-corrosion synergy was recently achieved via dense ultrafine twins[26], the issues remain that either the overall properties are far from satisfaction[23,24] or the fabrication process is complicated[26]. It is worth mentioning that the corrosion rate of Mg-RE binary alloys could be reduced by the addition of a third alloying element accompanied with microstructure optimization. It was found that the introduction of long-period stacking ordered (LPSO) phase can change the corrosion behavior of Mg-RE alloys from pitting corrosion to uniform corrosion and reduce the corrosion rate to some extent[27]. Zinc is a commonly added third element in Mg-RE alloys to build LPSO phase via conventional casting and subsequent thermal deformation. However, there are still clear gaps between the overall properties of Mg-RE-Zn alloys and the requirements in service conditions[6,22].

In this work, we propose a Mg-Y-Al system containing LPSO phase as a promising candidate of next-generation high performance Mg alloys based on the following two hypotheses. Firstly, Al addition into Mg-Y based alloys can form stable $Al_2Y$ phase in molten magnesium, which provides effective heterogeneous nucleation sites to promote the formation of fine and equiaxed α-Mg grains during solidification, which could benefit mechanical properties[28]. Secondly, the acceleration effect of corrosion rate by adding Al is less impressive than that of Zn[16], making it more easily deposited on the surface of magnesium than Zn in a neutral solution after dissolution[29,30]. Then, we successfully fabricate a model alloy (Mg-11Y-1Al, wt. %) through conventional casting followed by extrusion. The microstructure at different length scales, mechanical and corrosion behavior of Mg-11Y-1Al alloy and its base alloy Mg-11Y are characterized comprehensively. The underlying mechanisms of its strength, ductility and corrosion resistance are also discussed, respectively.

## Results and discussion
### Fabrication of high-performance Mg-11Y-1Al alloy
Figure 1a shows the corrosion resistance of commercial pure magnesium (CP-Mg, ≤ 99.95 wt. %), AZ91D, Mg-11Y (W11) alloy, Mg-11Y-1Zn-0.35Zr (WZ111K) alloy and Mg-11Y-1Al (WA111) alloy in various states (see the meaning of relevant designations in Table 1). Although HP-Mg is supposed to have a reasonably good corrosion resistance, the weight loss rate of CP-Mg is as high as 3.00 mg cm$^{-2}$ day$^{-1}$ (equivalent to 6.26 mm y$^{-1}$) due to the existence of impurities. The corrosion resistance of W11 alloy is even worse than the CP-Mg (as shown in Fig. 1a), which is generally attributed to the strong galvanic corrosion caused by the large potential difference between β phase ($Mg_{24}Y_5$) and magnesium matrix. Adding Zn into W11 alloy alleviates the corrosion rate to some extent and makes WZ111K alloy comparable with CP-Mg and AZ91D, but the weight loss rate is still higher than 1.00 mg cm$^{-2}$ day$^{-1}$ (2.09 mm y$^{-1}$). In contrast, the addition of 1 wt. % Al in W11 alloy improves the corrosion resistance dramatically (0.12 mg cm$^{-2}$ day$^{-1}$, equivalent to 0.25 mm y$^{-1}$). The weight loss rate was reduced by two orders of magnitude compared to W11 alloy and reduced by one order of magnitude compared to WZ111K alloy. The microstructure of as-cast WA111 (Fig. 1c) features with an average grain size of ~40 μm (Supplementary Fig. 4b) credited to the in-situ formation of $Al_2Y$ and LPSO phases. During the subsequent solution treatment at 520 °C for 8 h

(Fig. 1d), there is no noticeable change of the grain size attributed to the pinning effect of LPSO and $Al_2Y$ particles along the grain boundaries. The corrosion rate of solution-treated sample is as good as the as-cast one (0.10 mg cm$^{-2}$ day$^{-1}$, equivalent to 0.21 mm y$^{-1}$). The average grain size of WA111 alloy was further refined to 2 μm after extrusion (Fig. 1e and Supplementary Fig. 4) credited to the dynamic recrystallization. However, the significant change of grain size after extrusion only induces a slight reduction of corrosion rate to 0.08 mg cm$^{-2}$ day$^{-1}$ (0.17 mm y$^{-1}$).

Apart from the change of grain size, the amount of LPSO phase also varies substantially in different states of WA111 alloy between 6.5 % and 17 %. However, the corrosion rate remains at a fairly constant level i.e. 0.16–0.25 mm y$^{-1}$ though a weak dependence was observed between the corrosion rate and the volume fraction of LPSO phase, as shown in Supplementary Figs. 1 and 2. Consequently, a key finding from Fig. 1 is that the excellent corrosion resistance of WA111 alloy shows low sensitivity to microstructure. Therefore, we can further optimize the mechanical properties of WA111 alloy through thermal-mechanical treatment without compromising its corrosion resistance. Supplementary Fig. 4a presents typical stress-strain curves upon tension of the T4-1 and T4-EX-1 WA111 samples. The yield strength of WA111 alloy after solution treatment at 520 °C for 8 h (T4-1) is 160 MPa. After extrusion at 350 °C, the yield strength was increased dramatically to 350 MPa (T4-EX-1), with an enhanced tensile elongation of 8%.

In general, the key factors contributing to the yield strength ($\sigma_{0.2}$) of a magnesium alloy mainly consist of the friction stress of pure Mg ($\sigma_{Mg}$), solid-solution strengthening ($\sigma_{ss}$), grain-refinement strengthening ($\sigma_{gb}$), precipitation strengthening ($\sigma_{ppt}$) as well as texture strengthening ($\sigma_{tex}$), which can be expressed via the following equation.

$$\sigma_{0.2} = \sigma_{Mg} + \sigma_{ss} + \sigma_{ppt} + \sigma_{gb} + \sigma_{tex} \qquad (1)$$

After extrusion, the yield strength increment of the T4-EX-1 WA111 sample is about 190 MPa. Since there is neither noticeable difference of solute concentration nor the volume fraction of secondary phases between T4 and T4-EX samples, the yield strength increment ($\triangle\sigma_{0.2}$) after extrusion was probably contributed to the grain refinement ($\triangle\sigma_{gb}$) as well as texture strengthening ($\triangle\sigma_{tex}$). In this work, a modified Hall-Petch relationship below is usually used to describe the compounded effect of grain size and texture on the yield strength[24,31],

$$\sigma_{y(gb+tex)} = \frac{0.3}{m_t}\left(\sigma_0 + kd^{-1/2}\right) \qquad (2)$$

where $m_t$ and $k$ are the average Schmid factor and the Hall-Petch slope, respectively while $\sigma_0 = \sigma_{Mg} + \sigma_{ss} + \sigma_{ppt}$. The average grain sizes in the T4-1 and T4-EX-1 samples are 40 μm and 2 μm (Supplementary Fig. 4), respectively. The average Schmid factor $m_t$ of T4-EX-1 sample is about 0.27 (Supplementary Fig. 5), and the Hall-Petch factor $k$ is set as 252 MPa·μm[32,33] (Supplementary Note 2). Therefore, the compounded contribution of grain refinement and texture to the yield strength increment can be quantified to be 170 MPa based on Eq. (2), with $\triangle\sigma_{gb} = 135$ MPa $and$ $\triangle\sigma_{tex} = 35$ MPa, respectively. The rest of yield strength increment (20 MPa out of 190 MPa) might be attributed to the residual forest dislocations because immediate water-quenching just after extrusion could prohibit some dislocations from recovery (as shown in Supplementary Fig. 6). On the other hand, some LPSO lamellae were bent away from basal plane and hence increased the hindrance of basal dislocations (also shown in Supplementary Fig. 6). These two factors could further enhance the yield strength by 20 MPa to 350 MPa.

The reasons for achieving a moderate elongation after extrusion (e.g. from 4% of the T4-1 to 8% of the T4-EX-1) are threefold. Firstly, the reduction of grain size is beneficial on the ductility of magnesium[11].

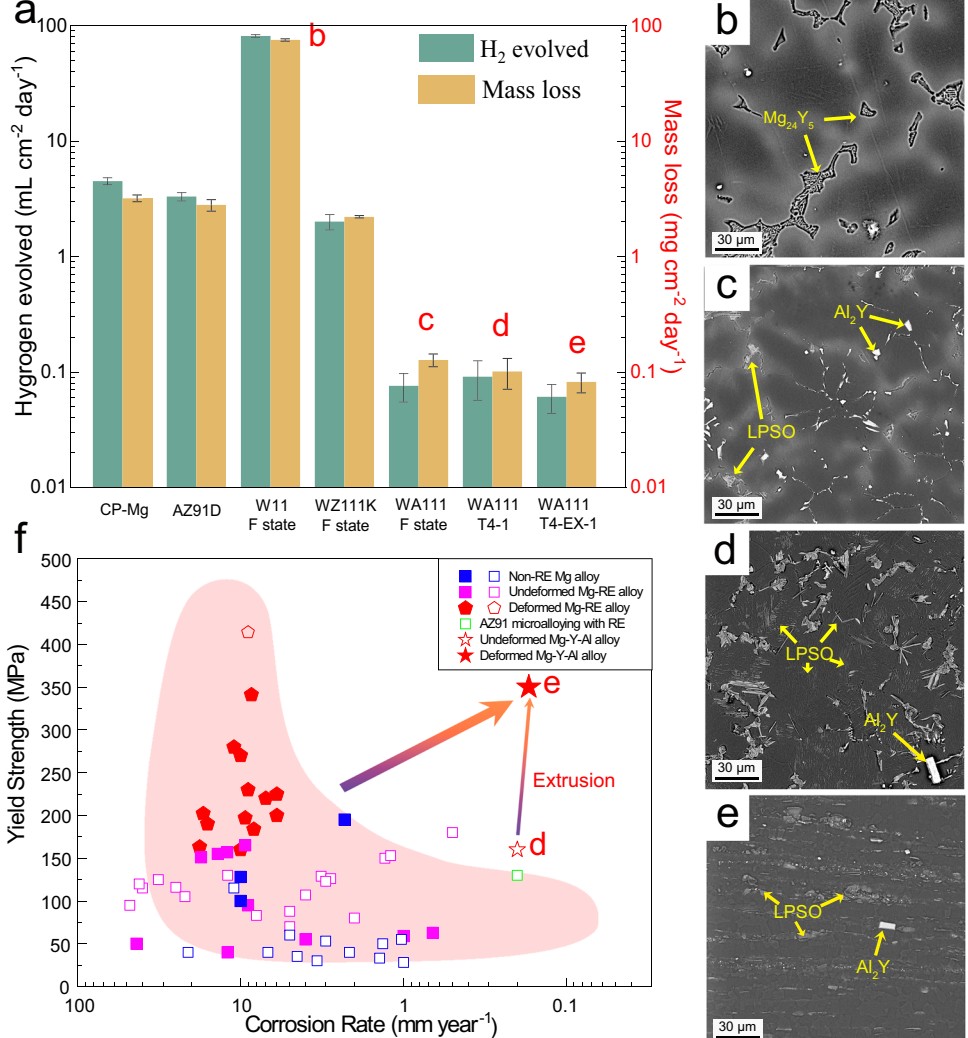

**Fig. 1 | The combination of mechanical and corrosion resistant properties of WA111 alloy shows competitive advantage over other comparative magnesium alloys in the literature. a** Hydrogen evolution and weight loss rates; **b** microstructure of as-cast (F) W11 sample; **c–e** microstructures of as-cast (F), T4-1 and T4-EX-1 WA111 samples, respectively; **f** comparison of yield strength and corrosion rate between the designed WA111 alloy and other magnesium alloys prepared by conventional methods, where alloys with elongation lower than 5% are denoted by open symbols while those higher than 5% are denoted by full symbols, and the oval backgrounds with pink and blue represent general tendency for strength-corrosion trade-off, respectively. (The corrosion rate was converted from the data obtained from immersion experiment in 3.5 wt. % NaCl solution). (References of each data point are listed in Supplementary Fig. 17).

Secondly, the severe stress concentration around Al₂Y particles in WA111 alloy is significantly alleviated by the grain refinement caused by the extrusion process, thus delaying the fracture of the alloy[34]. Last but not least, sufficient non-basal slip systems are activated in the extruded WA111 alloy (See Supplementary Fig. 7). Such a model alloy with a corrosion rate lower than 0.20 mm y⁻¹ with high yield strength of 350 MPa and moderate elongation of 8% shows competitive advantage over other comparative magnesium alloys in the literature, as shown in Fig. 1f.

**Table. 1 | Several states of WA111 alloy and corresponding designations**

| States of WA111 alloy | Designations |
| --- | --- |
| 520 °C-8 h | T4-1 |
| 550 °C-16 h | T4-2 |
| 520 °C-8 h-extruded at 350 °C | T4-EX-1 |
| 550 °C-16 h-extruded at 350 °C | T4-EX-2 |

### Anti-corrosion mechanism in Mg-11Y-1Al alloy

In addition to the unusual high corrosion resistance that shows low sensitivity to microstructure, the WA111 alloy also features with uniform corrosion behavior during salt spray and soaking experiments (Supplementary Figs. 8 and 9). Furthermore, the surface integrity of the WA111 alloy can be well maintained after being soaked in 3.5 wt. % NaCl solution for 14 days. In order to find out the underlying mechanism of low corrosion rate and uniform corrosion behavior of the WA111 alloy, the surface corrosion product of W11 alloy and WA111 alloy was carefully characterized, as shown in Fig. 2. For W11 alloy, the corrosion product is mainly comprised of Mg, Y and O elements. There is a corrosion product film growing rapidly on the surface and the local growth rate varies markedly from one site to another. For example, after being soaked for 1 day, its average film thickness reaches ~50 μm, but the thickness varies in the range of 20–100 μm at different sites (Fig. 2a). In contrast, the film of WA111 alloy soaked for 1 day is only about 5 μm thick (Fig. 2b), and its thickness is very uniform across the whole surface. This thin and uniform corrosion product also contains Mg, Y and O elements while the Al element is hard to be detected with EDS in such a scale. When the immersion

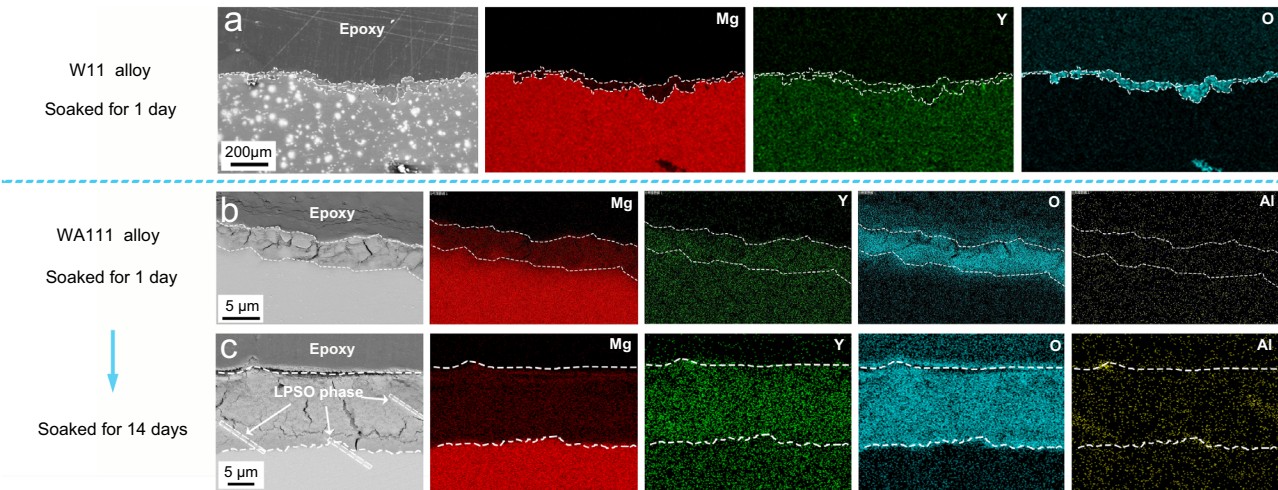

**Fig. 2 | Morphology and element distribution of corrosion product films in the W11 and WA111 alloys, respectively. a** As-cast W11 binary alloy soaked for 1 day; **b** as-cast WA111 alloy soaked for 1 day; **c** as-cast WA111 alloy soaked for 14 days.

time of WA111 alloy is extended to 14 days, the film thickens to 20-40 µm (Fig. 2c), which is still relatively uniform and much thinner than that of the W11 alloy soaked for 1 day. The EDS results of Fig. 2c show that the Y concentration in the film of WA111 alloy is higher than that in the matrix, and trace Al element in the film is detectable as well. The collected growth rate of corrosion product between W11 and WA111 alloys is also consistent with the hydrogen evolution data shown in Fig. 1a. Therefore, such a drastic change in corrosion behavior of the WA111 alloy should be credited to the addition of 1 wt. % aluminum.

A follow-up question is how aluminum addition improves the corrosion resistance of Mg-11Y alloy. After clearing the surface corrosion product, it was found that $Al_2Y$ particles and magnesium matrix were well protected while the LPSO phase disappeared (Supplementary Fig. 10). This seems to indicate that the LPSO phase is preferentially corroded and may be the anode phase. However, the LPSO phase was found to be a cathode phase based on our potential measurement data supported in Supplementary Fig. 3. The disappearance of LPSO phase during immersion test is attributed to that the interface between the LPSO phase and the Mg matrix was electrochemically corroded firstly and the LPSO phase peeled off during the cleaning process. The cathode role of LPSO phase is also shown in Fig. 2c. The embedded LPSO phase in the film indicates that they were protected well by the matrix due to the galvanic effect. It is worth mentioning that the $Al_2Y$ phase, with a higher potential than the Mg matrix (Supplementary Fig. 3), is a cathode phase as well. Therefore, we can rule out the possibility of anode protection mechanism and the actual contributor of WA111 alloy should be the protective film per se.

The key characteristic of such a protective film in the WA111 alloy is that: the film is thin and dense, which covers the surface evenly. It is worth mentioning that the content of yttrium in the film is much lower than that in the matrix of W11 alloy (Fig. 2a and Supplementary Fig. 11a) while the content of yttrium in the film is much higher than that in the matrix of WA111 alloy (Fig. 2c and Supplementary Fig. 11b). Based on EPMA analysis in Supplementary Fig. 11, the high concentration of Mg and O indicates that the corrosion product is mainly MgO in W11 alloy with a small amount of $Y_2O_3$. In contrast, the $Y_2O_3$ becomes the major constituent with some MgO and a small amount of $Al_2O_3$ in the film of WA111 alloy. In addition, TEM (Supplementary Fig. 12) and XRD (Supplementary Fig. 13) characterization were also used to further validate the phase identity in the film, which match well with the lattice constants of $Y_2O_3$, confirming that $Y_2O_3$ is the major constituent in the protective inner layer of WA111 alloy.

In order to have an in-depth understanding of the corrosion behavior of WA111 alloy, XPS analysis was also conducted to identify the surface chemistry of the film. Figure 3 presents XPS results of the film in the as-cast WA111 alloy. As shown in Fig. 3a, the concentration of Mg and O nearly remains unchanged with the increase of etching time from 0 to 8000 s, while the concentration of Y slowly rises. This part should be the outer MgO layer. Between the etching time from 8000 to 14000 s, the concentration of Mg gradually decreases, and the concentration of Y increases substantially. The presence of Y and Al elements indicates that this part should be the inner layer. In the region of etching time longer than 14000 s, the concentration of O declines sharply with the concentration of Mg increasing and the concentration of Y and Al remaining stable, which should be close to matrix. This XPS result confirms that there is a large amount of Y in the inner layer. In Fig. 3b–d, we can see the valence states of Mg, Y and Al in the corrosion product, confirming that they exist in the form of MgO, $Y_2O_3$ and $Al_2O_3$, respectively. And these oxides are probably transformed from corresponding hydroxides (i.e. $Mg(OH)_2$, $Y(OH)_3$ and $Al(OH)_3$) during soaking, respectively.

The electrochemical properties between W11 and WA111 alloys are compared in Fig. 4. Figure 4a shows that the corrosion potential of W11 alloy is more positive than that of WA111 alloy, indicating a small amount addition of Al reduces the corrosion potential significantly. The mixed potential theory indicates that such a significant reduction is attributed to the strong suppression of the cathodic hydrogen evolution reaction[35], which is also confirmed by the hydrogen evolution volume in Fig. 1a. Figure 4b shows that the corrosion current density of WA111 alloy is much lower than that of W11 alloy. This is consistent with the corrosion rates in Fig. 1a. According to the EIS curves of W11 and WA111 alloys (Fig. 4b) and the fitting data in Supplementary Table 1 and Supplementary Fig. 14, the film resistance $R_f$ of WA111 alloy in any state is two orders of magnitude higher than that of W11 alloy, indicating a more protective film formed on the surface of WA111 alloy. The variation of the $R_f$ value at different immersion time indicates that the film keeps growing in the first three days, which continually improves the protection effect for the matrix (Fig. 4c). However, the corrosion resistance of the film reaches a plateau after being soaked for 3 days. Combined with the measured film thickness in Fig. 2, it appears that the film starts protecting the underneath Mg alloy at the early stage, but the best protective effect can be achieved when the film is thicker than 10 µm. The thickness evolution of the corrosion product film (Supplementary Fig. 15b) shows that the growth rate of the film slowed down with immersion time, which is consistent with

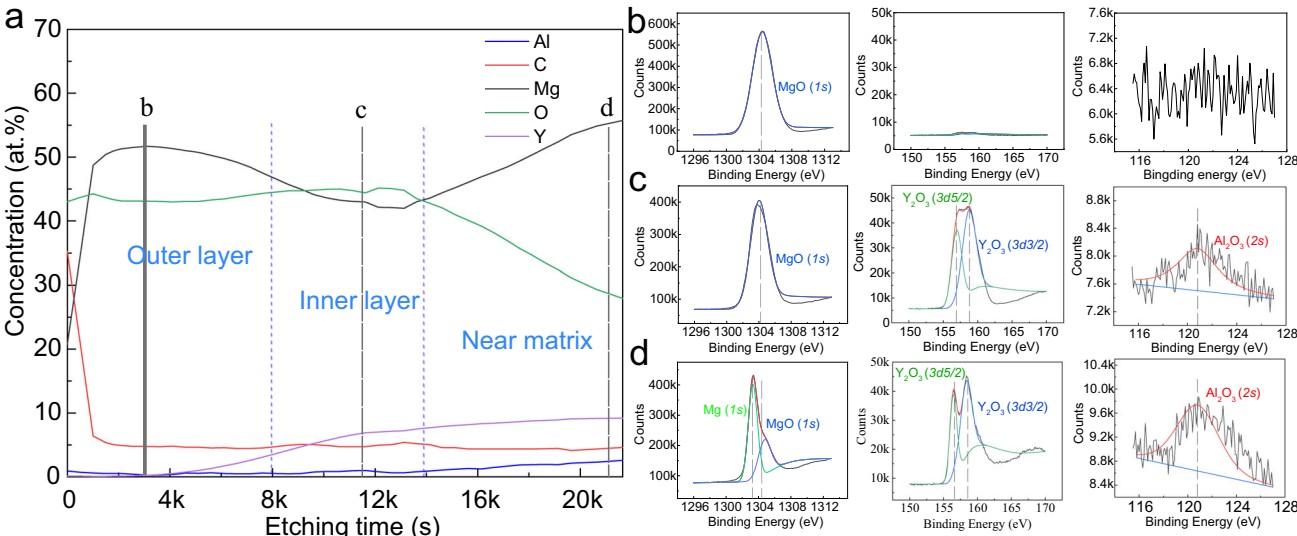

**Fig. 3 | Surface chemistry of the corrosion product film formed on the as-cast WA111 alloy. a** XPS depth profile, indicating various surface layers present (XPS etching rate, 0.08 nm s$^{-1}$ for Ta$_2$O$_5$); **b-d** are the corresponding element valence state information of Mg, Y and Al in the position of **b-d** in **a**, confirming that they exist in the form of MgO, Y$_2$O$_3$ and Al$_2$O$_3$, respectively.

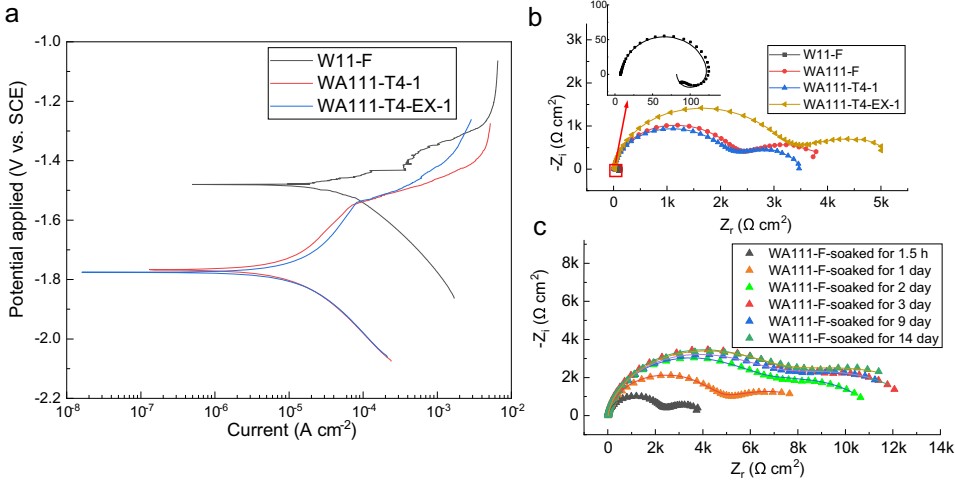

**Fig. 4 | Comparison of electrochemical properties between W11 and WA111 alloys. a** Potentiodynamic polarization (PDP) curves for W11 and WA111 alloys after soaking for 1.5 h; **b** Nyquist plots from electrochemical impedance spectroscopy (EIS) for W11 and WA111 alloys after soaking for 1.5 h; **c** Nyquist plots from EIS for the as-cast (F) WA111 alloy after soaking for different times. The fitting lines of these Nyquist plots are also listed above.

the hydrogen evolution curves (Supplementary Fig. 15a). Therefore, further thickening of the film after early stage has little effect on the corrosion rate of WA111 alloy.

Figure 5 summarizes the corrosion resistance mechanism of WA111 alloy against W11 alloy based on the above analysis. For W11 alloy, on one hand, the loosely packed MgO/Mg(OH)$_2$ film could not protect the Mg alloy effectively. On the other hand, the potential of the second phase is higher than that of the surrounding matrix, which could cause serious galvanic corrosion. As a consequence, some iso-lated corrosion products are formed in the early immersion stage of W11 alloy (Fig. 5a). With the extension of immersion time, the corrosion products cover the alloy surface loosely and even a very thick film can not provide good protection for the W11 alloy (Fig. 5c). However, the corrosion behavior of ternary WA111 alloy is totally different. When it is exposed to the brine solution, some solute yttrium that initially dis-solve with the Mg matrix forms an inner Y$_2$O$_3$/Y(OH)$_3$ layer deposited on the surface and protects the alloy underneath which is summarized

in Fig. 5b. The fast deposition of yttrium is mainly credited to the addition of aluminum while the yttrium alone is ineffective. The reason is probable that the solubility of aluminum in the brine solution is very small and hence it prefers to deposit on the alloy surface even at a very low concentration. When the Al and Y solute atoms on the sample surface are in contact with the brine solution, they lost electrons and then transform into Al$^{3+}$ and Y$^{3+}$ ions, respectively, which dissolve into the solution simultaneously. In contrast, the Al and Y atoms in the second phases do not dissolve into the solution because the second phases containing Al and Y are cathode phases protected by the matrix, as shown in Supplementary Fig. 3. Due to the low solubility of Al$^{3+}$ and Y$^{3+}$ (According to the solubility product constants (K$_{sp}$) of various ions in solution, the saturated solubility of ions at a certain pH value can be obtained[36]. When a magnesium alloy is soaked in brine, the pH value of its surface is about 10.3[37,38] and the approximate saturated solubilities of Al$^{3+}$ and Y$^{3+}$ ions are about 1.33 × 10$^{-22}$ mol L$^{-1}$ and 8.19 × 10$^{-12}$ mol L$^{-1}$, respectively.), their concentrations in the brine

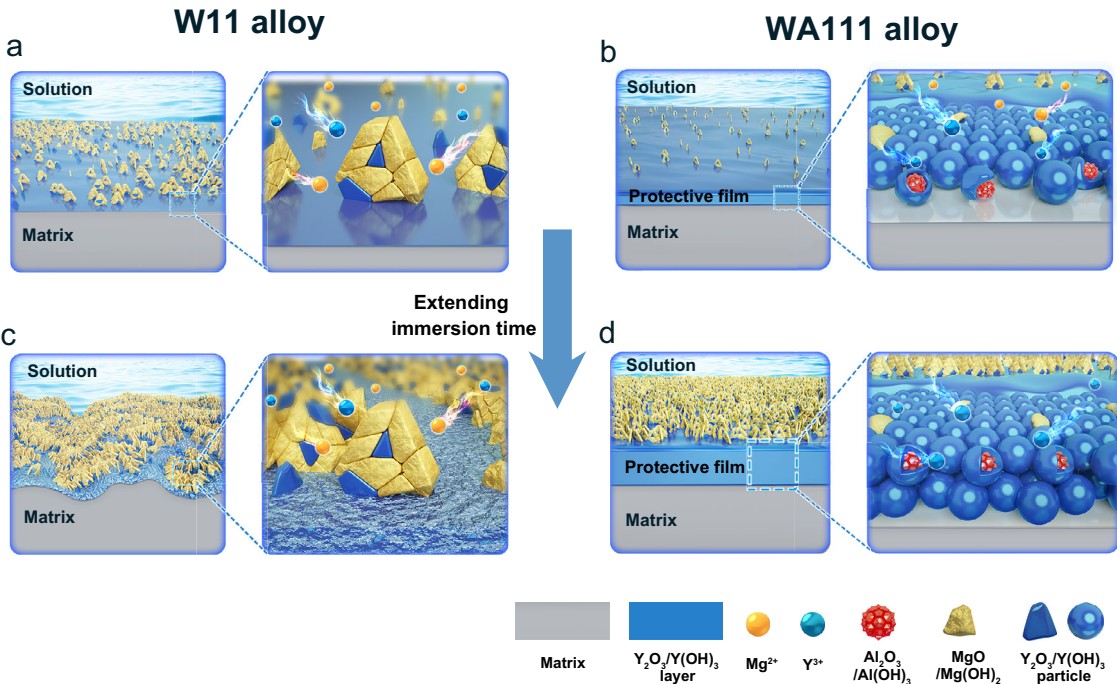

**Fig. 5 | Schematic diagram of corrosion resistance mechanism of W11 and WA111 alloys.** Early stage in brine solution of **a** W11 and **b** WA111 alloy; long-time exposure in brine solution of **c** W11 and **d** WA111 alloy.

solution quickly reach saturation, leading to the precipitation of Al and Y hydroxides on the alloy surface and the subsequent formation of protective product film. These initial deposition products (most likely $Al(OH)_3$) provide nucleation sites for subsequent deposition of other corrosion products, like $Y_2O_3/Y(OH)_3$ and $MgO/Mg(OH)_2$. Therefore, the deposition rate of $Y_2O_3/Y(OH)_3$ in WA111 alloy is much faster than that of W11 alloy, which promotes the formation of a protective layer on the surface of WA111 alloy as shown in the Fig. 5d. This hypothesis is also validated by the morphology of corrosion products on the surface of W11 and WA111 alloys after different soaking times. Supplementary Fig. 16 indicates that the formation of film layer on the surface of WA111 alloy is significantly faster than that of W11 alloy and the EIS test in Fig. 4c indicates the film on the surface of WA111 alloy is very protective. This protective film strongly inhibits the galvanic corrosion between second phases and Mg matrix. As a result, the WA111 shows low corrosion sensitivity to the microstructure.

As aforementioned, it is a long-standing challenge to improve the mechanical properties and corrosion resistance of Mg alloys simultaneously. The WA111 alloy developed in our current study delivers a desirable solution towards this target. The combination of excellent corrosion resistance and yield strength of WA111 alloy makes itself a superior candidate for wider demanding applications. In addition, it is also promising to extend the current alloy design strategy to other Mg-RE-Al alloy systems to develop a series of strong and corrosion resistant Mg alloys.

## Methods

### Casting and extrusion

Magnesium alloys with composition (wt. %) of Mg-11Y (W11), Mg-11Y-1Al (WA111) and Mg-11Y-1Zn-0.35Zr (WZ111K, the addition of 0.35Zr here aims to obtain a close average grain size with the as-cast WA111 alloy) were cast into a cylindrical mild steel mould with a diameter of 60 mm and a length of 200 mm. The alloys were melted in a protective gas of dry mixed $CO_2$ and $SF_6$ at 750 °C followed by isothermally held for 30 min before casting. The chemical compositions of W11 and WA111 alloys determined by Inductive Coupled Plasma (ICP) Emission Spectrometer are listed in Supplementary Table. 2. The ingots of WA111 alloy were solution treated at 520 °C for 8 h (denoted as T4-1) and 550 °C for 16 h (denoted as T4-2) followed by water quenching at room temperature, respectively. The solution-treated samples were extruded from a cylindrical ingot into a 15 mm diameter bar with an extrusion ratio of 16:1 at 350 °C followed by water quenching (denoted as T4-EX-1 and T4-EX-2, respectively).

### Microstructure characterization

Samples were cut from W11 and WA111 alloys in different processed conditions, and examined by optical microscope (OM) using a Zeiss Axioscope 5 machine. Images of Scanning Electron Microscope (SEM), Electron Back-Scattered Diffraction (EBSD) and Energy dispersive X-ray Spectroscopy (EDS) analyses were conducted in a Nova-FEI-230 machine. TEM specimens of WA111 alloy after tension test were cut into tablets of ~0.8 mm thick and further ground to a sheet of ~150 μm thick. Discs with 3 mm in diameter were punched from the sheet and further ground to ~70 μm thick, followed by twin-jet electro-polishing in an ethanol solution with 2 pct perchloric acid. Afterwards, the specimens were perforated in a Gatan 695 ion-milling system with 1.5 k eV for 20 min. The TEM lamella of WA111 alloy after corrosion was prepared using a Tescan Lyra-3 Focused Ion Beam (FIB) - SEM dual beam microscope with a Ga+ ion beam operated at 30 kV. The final voltage of 5 kV and final current of 50 pA were used to minimize the damage induced by the FIB milling. Bright field TEM images were taken using a JEOL-2100F TEM operated at 200 kV. STEM images and EDS mapping is collected in a FEI-Talos-F200X TEM operated at 200 kV.

### Chemical analysis

The compositions of the thin film and matrix were analyzed by electron probe microanalysis (EPMA) using Wavelength Dispersive Spectroscopy (WDS) with a voltage of 20 kV in an EPMA-8050G SEM. XPS (X-ray photoelectron spectroscopy) test was conducted by using a Thermo Scientific K-Alpha instrument with an Al-K$_\alpha$ X-ray source (monochromator) as anode with power of 105 W. Depth profiling was conducted by using argon ion sputtering with a 3 k eV energy.

## Tensile

The tensile test was conducted with dog-bone specimens with 5 mm in diameter and a gauge length of 30 mm. Every testing condition was repeated for three times.

## Corrosion tests

Hydrogen evolution and weight loss results were collected from an immersion test using samples with size of $1 \times 1 \times 0.5\ cm^3$ conducted in 3.5 wt. % NaCl solution at room temperature for two weeks. The hydrogen evolution rate ($mL\ cm^{-2}\ day^{-1}$) was converted into the weight loss rate ($mm\ year^{-1}$) via multiplying by a factor of 2.088 at 25 °C[12]. Continuous salt spray test was carried out by sample with size of $5 \times 5 \times 0.5\ cm^3$ using 5 wt. % NaCl solution at 35 °C in a salt spray chamber for two weeks. The NaCl solution is prepared by analytical grade NaCl and ultrapure water. At least three parallel samples were used for each test and the results were averaged. The samples were mechanical polished using 2000# SiC paper and cleaned with alcohol and dried by compressed air. The corrosion products of magnesium were cleaned by a cleaning agent composed of $Cr_2O_3$, $AgNO_3$, $Ba(NO_3)$ and ultrapure water for 3 min.

## Electrochemical analysis

Electrochemical test was carried out at room temperature in 3.5 wt. % NaCl solution. A magnesium electrode, a platinum sheet (specific area is $1 \times 1\ cm^2$), and a saturated calomel electrode (SCE) were used as working electrode, counter electrode, and reference electrode, respectively. These samples were embedded with epoxy resin and the exposed area is $1 \times 1\ cm^2$. Potentiodynamic polarization (PDP) experiments were conducted from $E_{OCP} - 0.4\ V$ to $E_{OCP} + 0.4\ V$ with a scan rate of $0.5\ mV\ s^{-1}$ to study the corrosion behavior of the WA111 alloy in particular its corrosion potential ($E_{corr}$) and its corrosion current ($i_{corr}$)[39]. Electrochemical impedance spectroscopy (EIS) was also carried out to assess the electrochemical response as a function of immersion time. The EIS was measured at OCP (Open Circuit Potential) in the frequency range from 100 kHz to 10 mHz with an AC amplitude of 10 mV.

## Error bars

The hydrogen evolution, weight loss and corrosion film thickness of the alloy were calculated as the average of three samples, and the error bars were plotted according to their standard deviation. Five points of EPMA analysis in each alloy were tested and the results were averaged, and the errors are calculated according to their standard deviation.

# Data availability

All relevant data supporting the findings of this study are contained in the paper and its supplementary information file. All other relevant data are available from the corresponding author on request.

# Code availability

All codes are included in the paper.

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

## Acknowledgements

This work was financially supported by National Natural Science Foundation of China (Nos. 51825101, 51701121 and 52001199), Shanghai Sailing Program (No. 17YF1408800), Science and Technology Commission of Shanghai Municipality (No. 18511109302), and Startup Fund for Youngman Research at SJTU (No. 18X100040022).

## Author contributions

D.Q., Y.L., Q.Z. and X.Z. conceived the idea. Q.Z., Y.L., F.C. and D.Q. wrote the manuscript. Q.Z. and H.Z. prepared the alloys. Q.Z. performed the mechanical tests, corrosion experiments and microstructure characterization. J.W. was responsible for the EPMA test. F.C. and Y.Y. were contributed to EIS analysis and anti-corrosion mechanism. Y.T. and W.D. participated in discussions and all authors contributed to the improvement of the paper.

## Competing interests

The authors declare no competing interests.
