## [Peer Review File · Nature Communications]

Title: Towards development of a high-strength stainless Mg alloy with Al-assisted growth of passive filmREVIEWER COMMENTS

Reviewer #1 (Remarks to the Author):

The author states that the goal of this paper is to develop a high-strength stainless magnesium alloy and to understand the mechanisms for high strength and high corrosion resistance.

For this, more clear explanations and answers for the following are required.

1. Line 104: It is worth mentioning that the as-extruded Mg-11Y-1Al exhibits much higher yield strength up to 350 MPa and also enhanced tensile elongation of 8% (See Fig. S1a). The strength increment is mainly attributed to the very fine grain size (2-5 μm) while the reasons of ductility enhancement are threefold.

There could be other factors contributing to this strength increment of 190MPa. In order to make above statement, the authors should clearly analyze and prove that the strength increment of 190MPa is mainly come from the grain refinement by using appropriate Hall-Petch equation reported for Mg alloys.

2. Line 109: Secondly, a weak texture after extrusion (as shown in Fig. S1b) also contributes to a higher elongation.

The texture of the extruded sample does not show significantly weak texture compared with that of as-cast sample. What makes you think the extruded sample shows weaker texture than the one before extrusion?

3. Line 101: A key finding from Fig. 1 is that the excellent corrosion resistance of Mg-11Y-1Al alloy is not sensitive to microstructure.

This is different from the viewpoints of majority of researchers in this field. (see review papers on corrosion: Ref. 4) In order to make this statement, a thorough study on microstructure, amounts of second phases, volta-potential differences between second phases and matrix, etc. should be thoroughly examined and analyzed for their contribution to the corrosion rate.

4. Line 228: The fast deposition of yttrium is mainly credited to the addition of aluminum while the yttrium alone is ineffective. The reason is that the solubility of aluminum in the salt solution is very small and hence it will prefer to deposit on the alloy surface even at a very low concentration.

The author argues that the reason why Mg-11Y-1Al alloy exhibits high corrosion resistance is as follows. First, Al ions with very low solubility in the salt solution rapidly form initial deposition products (most likely $\text{Al}(\text{OH})_3$) on the alloy surface, and these provide nucleation sites for other corrosion products, like $\text{Y}_2\text{O}_3/\text{Y}(\text{OH})_3$. Have they carried out any other more direct experiments to prove their hypothesis? If they have tried, what are the results?

5. Line 251: Magnesium alloys with composition (wt. %) of Mg-11Y, Mg-11Y-1Al and Mg-11Y-1Zn-0.35Zr were cast ... -> Why Mg-11Y-1Zn-0.35Zr?

6. Line 258: The deformed sample was extruded from a cylindrical ingot into a 15 mm diameter bar with

an extrusion ratio of 16:1 at 375°C followed by water quenching.
In Fig. S1 (a) 520°C-8h-350-16:1 Extrusion temperature and ratio?
Which one is the correct extrusion temperature, 350 °C or 375 °C?

Supplementary Materials

4. Microscopic surface morphology of Mg-11Y-1Al alloy after cleaning the corrosion products out of the immersion test in 3.5 wt.% NaCl solution for 14 days.

Fig. S4 shows the microscopic surface morphology of Mg-11Y-1Al alloys at different states after clearing the corrosion product out of the brine immersion experiment. In general, no serious corrosion was observed on the surface no matter what thermomechanical treatment has applied to the Mg-11Y-1Al alloys, while the integrity of the Mg-11Y-1Al alloy solution treated at 570 °C for 16 hour is slightly better than other states (Fig. S2c). This may be attributed to the change of volume fraction and distribution of LPSO phase in Mg-11Y-1Al alloy. In addition, comparing the microstructure Mg-11Y-1Al before and after immersion of the as-cast Mg-11Y-1Al sample (Fig. S2e vs. Fig. S2f), we found that the phase that peeled off first in the immersion test was mainly LPSO phase sitting along the grain boundaries.

the integrity of the Mg-11Y-1Al alloy solution treated at 570 °C for 16 hour (520 °C for 8 hour?) is slightly better than other states (Fig. S2c?).

In addition, comparing the microstructure Mg-11Y-1Al before and after immersion of the as-cast Mg-11Y-1Al sample (Fig. S2e vs. Fig. S2f) -> Fig. S4d vs. Fig. S4e?

Reviewer #2 (Remarks to the Author):

Developing a high-strength Mg alloy with excellent corrosion-resistance property has been a long-standing problem in industry. This work proposes a new alloy-design strategy, which obviously improve the corrosion-resistance in a high-strength Mg-Y-Al system and the research work is of great significance to the development of high corrosion resistant magnesium alloys. The manuscript is well organized, and it is recommended to accept the publication of the revised paper. Suggestions for minor modification are as follows.

1. Fig.1 shows the formation of LPSO phase in the Mg-Y-Al alloys. As a main second phase in this alloy, the authors did not discuss its effect on the corrosion performance. Please give a brief discussion on the effect of LPSO phase on corrosion performance.
2. The addition of Al seems to be crucial for the corrosion performance of the alloy system. What is the effect of the change in the Al/Y ratio on the corrosion resistance?
3. The hydrogen evolution volume against immersion time is necessary to illustrate the corrosion performance of Mg-Y-Al alloys.
4. The poor corrosion resistance of Mg-Y binary alloy is closely related to the existence of second phase, which can be adjusted by heat treatment. However, the solid solution treatment seems to have little influence on the corrosion performance of Mg-Y-Al alloys. Please explain this phenomenon in details.

5. The research work mainly shows the microstructure and properties of the alloy, but the related thermodynamics and kinetics with formation of corrosion product film are lack of analysis.
6. There are several important related papers in Journal of Magnesium and Alloys and Materials Science and Engineering. It is suggested to make a supplementary introduction in the introduction

Fusheng pan

Reviewer #3 (Remarks to the Author):

The corrosion behaviors of Mg-11Y-1Al (wt.%) in chlorine-containing environment were studied by conventional corrosion test and surface analysis. The mechanical and corrosion resistance of the alloy can be greatly improved by adding Al. The research of this manuscript is of high scientific significance and engineering application values. However, the authors do need to address the following concerns before it can be accepted for publication.

1. The authors use “stainless magnesium alloy” in title to emphasize the corrosion low rate. It is necessary to define “stainless” first. For example, under similar conditions, does the corrosion rate is comparable with “stainless steel” or not? A recently developed Mg-Ca alloy with the corrosion rate of 0.1mm/year is merely considered to approaching “stainless” (Materials Horizons, 2020). In addition, The authors need to cite and address the following two papers: “A high-specific-strength and corrosion-resistant magnesium alloy (Nature Materials, 2015)” and “Evading strength-corrosion tradeoff in Mg alloys via dense ultrafine twins (Nature Communications, 2021)”,
2. Only the yield strength and corrosion rate have been compared to the available statistical data, but the authors draw the conclusions that the combination of the corrosion rate, strength, and elongation of the Mg-11Y-1Al outperforms all the commercial magnesium alloys (Line11-13). The authors need to clarify the reason.
3. A $Y_2O_3/Y(OH)_3$ fast deposition model is proposed to illustrate the formation of the protective film. However, this model only considers the deposition process of the Mg-11Y and Mg-11Y-1Al, while the dissolve behaviors of Y^{3+} and Al^{3+} , which is equally important, are ignored.
4. The authors tries to interpret the moderate elongation which is rarely mentioned in the main text, while little explanation is given to the high strength which is mentioned in the title, and more likely to be a potential highlight.
5. It will be great if specific equation can be given to illustrate the effect of Al addition on the improvement of corrosion resistance.
6. Line 270, “K eV” should be “k eV”.
7. Line 273, space should be inserted between “50” and “pA”.
8. There is no testing parameter for PDP in the experimental section. What is the “standard PDP test”? How to choose the scanning potential range and scanning rate?
9. Is the platinum electrode sheet or mesh? What is the specific area?
10. What is the size of the electrochemical test sample? Is the potential applied by the EIS direct current or alternating current?
11. In immersion test section, the size of the sample and the immersion time are missing.

12. The Chinese parentheses in the in-depth analysis diagram should be changed to English.
13. The equivalent circuit and fitting line should be given in EIS.

Point by point response to reviewers' comments

Dear reviewers,

We sincerely appreciate the invaluable and constructive comments from you. The manuscript has been revised thoroughly to incorporate all the comments. In addition, six new figures have been included in the supplementary materials to address the suggestions provided by you. **The amendments were highlighted in red in the revised manuscript and the supplementary materials** (the order of figures in the supplementary materials were also adjusted to reflect the relevant changes in the main text), and a point-by-point response is enclosed below. We hope the revised version is now suitable for publication in *Nature communications*.

Yours Sincerely
Yangxin Li

Reviewer #1:

The author states that the goal of this paper is to develop a high-strength stainless magnesium alloy and to understand the mechanisms for high strength and high corrosion resistance. For this, more clear explanations and answers for the following are required:

Comment 1: Line 104: It is worth mentioning that the as-extruded Mg-11Y-1Al exhibits much higher yield strength up to 350 MPa and also enhanced tensile elongation of 8% (See Fig. S1a). The strength increment is mainly attributed to the very fine grain size (2-5 μm) while the reasons of ductility enhancement are threefold.

There could be other factors contributing to this strength increment of 190 MPa. In order to make above statement, the authors should clearly analyze and prove that the strength increment of 190 MPa is mainly come from the grain refinement by using appropriate Hall-Petch equation reported for Mg alloys.

[Response]: Thanks a lot for this helpful suggestion. In general, the key factors contributing to the yield strength ($\sigma_{0.2}$) of a magnesium alloy mainly consist of the friction stress of pure Mg (σ_{Mg}), solid solution strengthening (σ_{ss}), grain refinement strengthening (σ_{gb}), precipitation strengthening (σ_{ppt}) as well as texture strengthening (σ_{tex}), which can be expressed as the following equation.

$$\sigma_{0.2} = \sigma_{Mg} + \sigma_{ss} + \sigma_{ppt} + \sigma_{gb} + \sigma_{tex} \quad (1)$$

As shown in the **Fig. S4a**, the yield strength of T4 and T4-EX samples are 160 MPa and 350 MPa, respectively. Accordingly, the yield strength increases by 190 MPa after extrusion. It was found that there is neither noticeable difference of solute concentration nor the volume fraction of secondary phases between T4 and T4-EX samples, so the yield strength increment ($\Delta\sigma_{0.2}$) after extrusion should be attributed to the grain refinement strengthening ($\Delta\sigma_{gb}$) as well as texture strengthening ($\Delta\sigma_{tex}$). A modified

Hall-Petch relationship below is usually used to describe the compounded effect of grain size and texture on the yield strength^{1,2}:

$$\sigma_{y(gb+tex)} = \frac{0.3}{m_t} (\sigma_0 + kd^{-\frac{1}{2}}), \quad (2)$$

where m_t and k are the average Schmid factor and the Hall-Petch slope, respectively while $\sigma_0 = \sigma_{Mg} + \sigma_{ss} + \sigma_{ppt}$. For T4 sample, the Hall-Petch slope, k , is 250 MPa· $\mu\text{m}^{-1/2}$ ^{3,4}, and the average grain size is 40 μm while the texture strengthening is negligible ($m_t = 0.3$). The value of σ_0 , can then be back calculated to be about 120 MPa based on Eq. 2, given the yield strength of T4 sample is 160 MPa.

For the T4-EX sample, the average grain size decreases to 2 μm (**Fig. S4b**) and the average Schmid factor m_t reduces to 0.27 (**Fig. S5**). Suppose the value of σ_0 is unchanged, the value of $\sigma_{y(gb+tex)}$ turns out to be 330 MPa after extrusion based on Eq. 2. This indicates that the grain refinement and texture development accounts for the yield strength increment of (330 – 160) MPa = 170 MPa. The above analysis confirms that the grain refinement strengthening in conjunction with texture strengthening are the two main contributors of the yield strength increment (190 MPa) after extrusion.

The rest of yield strength increment (20 MPa out of 190 MPa) might be attributed to the residual forest dislocations because immediate water-quenching just after extrusion could prohibit some dislocations from recovery (as shown in **Fig. S6**). On the other hand, some LPSO lamellae were bent away from basal plane and hence increased the hindrance of basal dislocations (also shown in **Fig. S6**). These two factors could further enhance the yield strength by 20 MPa to 350 MPa.

It should also be noted that the description “The grain size of Mg-11Y-1Al alloy was further refined to 2-5 μm after extrusion” (Line 98, Page 5) in the original manuscript remains some ambiguity. In fact, we have undertaken extrusion at different temperatures. The grain size of 520°C-8 h solid solution state after extrusion at 350°C (T4-EX-1) is 2 μm while a slight increase of the grain size to 5 μm was observed in the 550°C-16 h solid solution state after extrusion at 350°C (T4-EX-2). For clarity, we only took grain size of 520°C-8 h solid solution state after extrusion at 350°C (T4-EX-1 sample) as an example in the revised manuscript.

Comment 2: Line 109: Secondly, a weak texture after extrusion (as shown in Fig. S1b) also contributes to a higher elongation. The texture of the extruded sample does not show significantly weak texture compared with that of as-cast sample. What makes you think the extruded sample shows weaker texture than the one before extrusion?

[Response]: We apologize for the ambiguity that remains in this sentence. We meant to say that the extruded Mg-11Y-1Al alloy (T4-EX-1 sample) exhibited weaker texture compared to that reported in most extruded Mg alloys in the literature. We have rephased the description in the revised manuscript to avoid any confusion.

Comment 3: Line 101: A key finding from Fig. 1 is that the excellent corrosion resistance of Mg-11Y-1Al alloy is not sensitive to microstructure. This is different from the viewpoints of majority of researchers in this field. (see review papers on corrosion: Ref. 4) In order to make this statement, a thorough study on microstructure, amounts of second phases, volta-potential differences between second phases and matrix, etc. should be thoroughly examined and analyzed for their contribution to the corrosion rate.

[Response]: We appreciate the reviewer's comment and agree that the original statement might be a bit misleading. It is undoubtful that the microstructure of magnesium alloys is usually a critical contributor to their corrosion resistance^{5,6}. For Mg-11Y-1Al alloy, polygonal Al₂Y and lamellar LPSO particles are the two major second phases that are present in the Mg-11Y-1Al alloy. As shown in **Fig. R1 (Fig. S3)** in the revised supplementary materials, the volta-potential of Al₂Y and LPSO phases are 300 mV and 400 mV higher than the surrounding matrix, respectively. The results indicate that these two second phases are both cathode phases. However, Al₂Y and LPSO phases only cause strong galvanic corrosion in the initial stage of the immersion test until a dense surface protective film fully developed in 1200 s, as shown in **Fig. S15**. During the following test, the galvanic corrosion between the second phases and the surrounding Mg matrix was effectively inhibited by this protective film. **Fig. R2 (Fig. S1)** in the revised supplementary materials shows five representative microstructures of Mg-11Y-1Al alloy after different thermal mechanical treatments. Although the amount and distribution of LPSO phase varies in these five different states, the corrosion rate remains at a fairly constant level i.e. 0.16-0.25 mm/y (**Fig. S2 and Fig. 1a**). That is why we concluded "the excellent corrosion resistance of Mg-11Y-1Al alloy is not sensitive to microstructure" in the original manuscript. Nevertheless, out of these five states, the Mg-11Y-1Al alloy with less amount of LPSO phase does show a slightly lower corrosion rate (**Figs. S1 and S2**). Therefore, we have rephrased this statement to "the excellent corrosion resistance of Mg-11Y-1Al alloy shows low sensitivity to microstructure due to the instant formation of the protective corrosion film" to reflect the effect of second phases volume fraction on the galvanic corrosion in the initial stage of immersion test.

Fig. R1 (S3) Potential of the (a-b) Al_2Y phase and (c-d) LPSO phase in the T4-1 state Mg-11Y-1Al alloy, respectively. The dotted lines in (b) and (d) represent the corresponding position of blue solid pentagon in (a) and (c), respectively.

Fig. R2 (S1). BSE images of Mg-11Y-1Al alloy after different thermal mechanical treatments. (a) F state; (b) T4-1 state; (c) T4-2 state, with the EDS results of marked Al_2Y and LPSO particles shown in (d); (e) T4-EX-1 state; (f) T4-EX-2 state.

Comment 4: Line 228: The fast deposition of yttrium is mainly credited to the addition of aluminum while the yttrium alone is ineffective. The reason is that the solubility of aluminum in the salt solution is very small and hence it will prefer to deposit on the alloy surface even at a very low concentration. The author argues that the reason why Mg-11Y-1Al alloy exhibits high corrosion resistance is as follows. First, Al ions with

very low solubility in the salt solution rapidly form initial deposition products (most likely $\text{Al}(\text{OH})_3$) on the alloy surface, and these provide nucleation sites for other corrosion products, like $\text{Y}_2\text{O}_3/\text{Y}(\text{OH})_3$. Have they carried out any other more direct experiments to prove their hypothesis? If they have tried, what are the results?

[Response]: We have carefully analyzed the surface morphology of the Mg-11Y and Mg-11Y-1Al alloys after soaking in the 3.5 wt. % NaCl solution with different time intervals. As shown in **Fig. S15**, scattered corrosion products on the surface of both Mg-11Y and Mg-11Y-1Al alloy were found around second phases due to the galvanic corrosion after soaking for 30 seconds. However, the corrosion products on the surface of Mg-11Y-1Al alloy developed much more quickly than that on the surface of Mg-11Y alloy. After soaking for 240 seconds, the surface of Mg-11Y-1Al alloy was almost fully covered by the corrosion products, while the surface of Mg-11Y alloy only showed isolated patches of corrosion products. After soaking for 1200 s, the surface of Mg-11Y-1Al alloy was fully covered by the corrosion products, while the surface of Mg-11Y alloy still showed isolated patches of corrosion products. After identifying the corrosion products of Mg-11Y and Mg-11Y-1Al alloys, as shown in **Figs. 2** and **3**, a much higher concentration of Y and a small amount of Al was discovered in the protective film of the Mg-11Y-1Al alloy. Such a finding indicates that it is highly likely that the addition of Al accelerates the deposition of $\text{Y}_2\text{O}_3/\text{Y}(\text{OH})_3$ and improves the corrosion resistance accordingly. It is worth mentioning that we have been looking for more direct evidence to validate our hypothesis, but it is rather difficult to design such an experiment to directly capture the nucleation process of $\text{Y}_2\text{O}_3/\text{Y}(\text{OH})_3$ film in the immersion test at this moment. We are looking forward to fully solving this problem through potential collaborations with research groups in chemistry community in a near future.

Comment 5: Line 251: Magnesium alloys with composition (wt. %) of Mg-11Y, Mg-11Y-1Al and Mg-11Y-1Zn-0.35Zr were cast ... -> Why Mg-11Y-1Zn-0.35Zr?

[Response]: The reasons of picking Mg-11Y-1Zn-0.25Zr in our study are as follows: (1) First of all, Zn is a commonly-used alloying element in magnesium alloys, which can help us compare its influence on corrosion-resistance with Al in the same Mg-11Y based system; (2) Secondly, the addition of Zn in Mg-Y system can induce the formation of LPSO phase, which can help us confirm whether the LPSO phase plays a crucial role in enhancing corrosion resistance; (3) Last but not least, the addition of Zr can reduce the average grain size of Mg-11Y alloy to a similar level as the Mg-11Y-1Al alloy. This helps identify whether grain refinement plays the predominant role in enhancing corrosion resistance. Given the similar level of LPSO phase and grain size, the much lower corrosion rate of Mg-11Y-1Al alloy over Mg-11Y-1Zn-0.25Zr should be credited to the Al addition, rather than the grain size or LPSO phase.

Comment 6: Line 258: The deformed sample was extruded from a cylindrical ingot into a 15 mm diameter bar with an extrusion ratio of 16:1 at 375 °C followed by water quenching. In Fig. S1 (a) 520°C-8h-350-16:1 Extrusion temperature and ratio?

Which one is the correct extrusion temperature, 350 °C or 375 °C?

[Response]: We apologize for this typo and it should be 350 °C. We have rectified this error in the revised manuscript.

Comment 7: Supplementary Materials 4. Microscopic surface morphology of Mg-11Y-1Al alloy after cleaning the corrosion products out of the immersion test in 3.5 wt.% NaCl solution for 14 days. Fig. S4 shows the microscopic surface morphology of Mg-11Y-1Al alloys at different states after clearing the corrosion product out of the brine immersion experiment. In general, no serious corrosion was observed on the surface no matter what thermomechanical treatment has applied to the Mg-11Y-1Al alloys, while the integrity of the Mg-11Y-1Al alloy solution treated at 570 °C for 16 hour is slightly better than other states (Fig. S2c). This may be attributed to the change of volume fraction and distribution of LPSO phase in Mg-11Y-1Al alloy. In addition, comparing the microstructure Mg-11Y-1Al before and after immersion of the as-cast Mg-11Y-1Al sample (Fig. S2e vs. Fig. S2f), we found that the phase that peeled off first in the immersion test was mainly LPSO phase sitting along the grain boundaries.

the integrity of the Mg-11Y-1Al alloy solution treated at 570 °C for 16 hour (520 °C for 8 hour?) is slightly better than other states (Fig. S2c?).

In addition, comparing the microstructure Mg-11Y-1Al before and after immersion of the as-cast Mg-11Y-1Al sample (Fig. S2e vs. Fig. S2f) -> Fig. S4d vs. Fig. S4e?

[Response]: We apologize for these typos in the original supplementary materials. For the first case, we meant to refer to the sample solution treated at 550°C for 16 hours, i.e. Sample T4-2, of which the appearance has been least affected after immersion test for 14 days in the undeformed states (**Fig. S9c** in the revised supplementary materials). For the second case, we meant to refer to Fig. S4d vs. Fig. S4e, which are **Fig. S10a** vs. **Fig. S10b** in the revised supplementary materials. All the above typos are now corrected in the revised version.

Reviewer #2:

Developing a high-strength Mg alloy with excellent corrosion-resistance property has been a long-standing problem in industry. This work proposes a new alloy-design strategy, which obviously improve the corrosion-resistance in a high-strength Mg-Y-Al system and the research work is of great significance to the development of high corrosion resistant magnesium alloys. The manuscript is well organized, and it is recommended to accept the publication of the revised paper. Suggestions for minor modification are as follows:

Comment 1: Fig.1 shows the formation of LPSO phase in the Mg-Y-Al alloys. As a main second phase in this alloy, the authors did not discuss its effect on the corrosion

performance. Please give a brief discussion on the effect of LPSO phase on corrosion performance.

[Response]: We appreciate the reviewer's suggestion to discuss the effect of LPSO phase on the corrosion performance. The volta-potential test (**Fig. S3**) shows that the LPSO phase has a potential 400 mV higher than the surrounding matrix. Hence, the presence of LPSO phase accelerates galvanic corrosion in the initial stage of the immersion test. As a result, the sample T4-2 (550°C-16 h) with a lower volume fraction of LPSO phase (6.5%) exhibited a slightly lower corrosion rate (0.19 mm/y) than that of sample T4-1 (520°C-8h, 0.21 mm/y) with a higher volume fraction of LPSO phase (17%), as shown in **Fig. S1 and S2**. However, the galvanic corrosion between the LPSO phase and Mg matrix is soon inhibited significantly through the formation of protective corrosion product film. Therefore, the corrosion rate shows a low sensitivity to the volume fraction LPSO phase in Mg-11Y-1Al alloy. Accordingly, we have added more descriptions in the revised manuscript (See Line 107-111, Page 5 as follows "Apart from the change of grain size, the amount of LPSO phase also varies substantially in different states of Mg-11Y-1Al alloy between 6.5% and 17%...").

Comment 2: The addition of Al seems to be crucial for the corrosion performance of the alloy system. What is the effect of the change in the Al/Y ratio on the corrosion resistance?

[Response]: We agree with the reviewer that the addition level of Al is crucial to maintain good corrosion resistance of this series of Mg-Y-Al alloys. In order to show the effect of the change in the Al/Y ratio on the corrosion rate of Mg-Y-Al alloy, we have manufactured quite a few additional cast Mg-Y-Al alloys with different Al and Y concentrations and investigated their corrosion performance. The corrosion rates of these Mg-Y-Al alloys were graphically illustrated in the bubble chart of **Fig. R3 (S17)** below, where the corrosion rate is proportional to the areas of the bubbles. The highest corrosion rates were observed from binary Mg-Y alloys (150 mm/y) and Mg-Al alloys (20 mm/y⁷). In contrast, the corrosion rate decreases dramatically when ternary Mg-Y-Al alloys are under consideration. The minimum corrosion rate (~ 0.2 mm/y) was found when Al concentration reaches 1 wt. %. Meanwhile, the variation of Y concentration between 4 wt. % and 11 wt. % does not induce noticeable change of the corrosion rate of Mg-Y-1Al alloys. In a nutshell, our current study indicates that the desirable corrosion rate (~ 0.2 mm/y) can be achieved in a wide range of Mg-(4-11)Y-1Al alloys. However, a higher Y addition is preferable due to the better mechanical properties. Accordingly, we have added the following sentences in the revised supplementary materials (See Line 304-321, Page 20-21 as follows "The corrosion rate of Mg-Y-Al alloys also depends on the addition level of Al and Y...").

Fig. R3 (S17) Effect of different Y and Al additions on corrosion rate of magnesium alloy. The size of the bubbles is proportional to the corrosion rate.

Comment 3: The hydrogen evolution volume against immersion time is necessary to illustrate the corrosion performance of Mg-Y-Al alloys.

[Response]: We appreciate the reviewer’s suggestion to illustrate the dependence of hydrogen evolution volume on the immersion time. The hydrogen evolution curves against immersion time of Mg-11Y-1Al alloy in different states can be seen in **Fig. R4a (S14a)** of the revised supplementary materials. It shows that the hydrogen evolution volume of the Mg-11Y-1Al alloy in different states all increases slowly with immersion time. However, the difference of the hydrogen volume between three specimens is marginal, indicating that the corrosion resistance of Mg-11Y-1Al alloy shows low sensitivity to microstructure due to formation of the protective film layer on the alloy surface. **Fig. R4b (S14b)** shows the thickness of the protective film layer with the immersion time. The thickness evolution of the corrosion product film (**Fig. S14b**) shows that the grow rate of the film slows down with immersion time, which is consistent with the hydrogen evolution curves (**Fig. S14a**). Accordingly, we have added the following sentences in the revised supplementary materials (See Line 235-256, Page 16-17 as follows “Fig. S14a shows hydrogen evolution curves against immersion time of three representative Mg-11Y-1Al samples...”).

Fig. R4 (S14) (a) Hydrogen evolution volume and (b) thickness evolution of corrosion product film against immersion time in the Mg-11Y-1Al alloy.

Comment 4: The poor corrosion resistance of Mg-Y binary alloy is closely related to the existence of second phase, which can be adjusted by heat treatment. However, the solid solution treatment seems to have little influence on the corrosion performance of Mg-Y-Al alloys. Please explain this phenomenon in details.

[Response]: We agree with the reviewer that for most Mg alloys, the corrosion resistance is closely related to the amount and distribution of second phases due to the micro-galvanic corrosion between the second phases and the surrounding Mg matrix. Therefore, solid solution treatment can affect the corrosion behavior of Mg alloys significantly through dissolving the second phases. The polygonal Al_2Y and lamellar LPSO are the two major second phases in Mg-11Y-1Al alloy. The volta-potential test (**Fig. S3**) shows that the Al_2Y and LPSO phase act as strong cathodes in the galvanic corrosion with Mg matrix. Consequently, the presence of Al_2Y and LPSO phases accelerates galvanic corrosion in the initial stage of the immersion test. However, such galvanic corrosion is soon inhibited significantly through the formation of protective corrosion product film, regardless of the heat treatment states of Mg-11Y-1Al alloy (**Fig. S9**). Therefore, the predominant factor of the corrosion behavior of Mg-11Y-1Al is the protective corrosion product film rather than the second phases.

Comment 5: The research work mainly shows the microstructure and properties of the alloy, but the related thermodynamics and kinetics with formation of corrosion product film are lack of analysis.

[Response]: We agree with the reviewer that the thermodynamics and kinetics of corrosion product film formation are of scientific significance to understand the unique corrosion behavior of this novel Mg-11Y-1Al alloy. Hence, we have undertaken additional studies on this topic in the last few months and the key results are presented as follows.

For thermodynamics, three Pourbaix diagrams are adopted to make an explanation, as shown in **Fig. R5**. In the early stage of the corrosion process, the alloy surface environment is slightly alkaline, which is beneficial for the deposition of Al and Y to form a protective corrosion product film.

For kinetics, the protective film formed quickly in the Mg-11Y-1Al alloy at the early stage as shown in **Fig. S15**, which is consistent with the polarization curves in **Fig. 3d**. The thickness evolution of the corrosion product film (**Fig. S14b**) shows that the growth rate of the film slows down with immersion time. More importantly, the EIS curves with different soaking time in **Fig. 3f** of the revised manuscript verify that the resistance of the Mg-11Y-1Al increases significantly in the first 1-2 days and reaches a plateau till the end. In summary, the protective corrosion product film of Mg-11Y-1Al forms instantly and soon covers the whole surface of the specimen in the immersion test. The corrosion resistance of film reaches a high level in 1-2 days and keeps stable till the end.

Fig. R5 Potential-pH equilibrium diagrams of (a) magnesium-water; (b) yttrium-water; (c) aluminum-water⁸

Comment 6: There are several important related papers in Journal of Magnesium and Alloys and Materials Science and Engineering. It is suggested to make a supplementary introduction in the introduction.

[Response]: We appreciate the reviewer's suggestion. We have reviewed more related papers and made a supplementary introduction in the revised manuscript. The new references included

[4] Paper 1 Atrens, A. *et al.* Review of Mg alloy corrosion rates. *Journal of Magnesium and Alloys* **8**, 989-998 (2020).

[5] Paper 2 Zhao, P., Ying, T., Cao, F., Zeng, X. & Ding, W. Designing strategy for corrosion-resistant Mg alloys based on film-free and film-covered models. *Journal of Magnesium and Alloys* (2021).

[8] Paper 3 Chen, T. *et al.* Coupling physics in machine learning to investigate the solution behavior of binary Mg alloys. *Journal of Magnesium and Alloys* (2021).

[23] Paper 4 Gerashi, E., Alizadeh, R. & Langdon, T. G. Effect of crystallographic texture and twinning on the corrosion behavior of Mg alloys: A review. *Journal of Magnesium and Alloys* (2021).

[24] Paper 5 He, J. *et al.* Enhancement of mechanical properties and corrosion resistance of magnesium alloy sheet by pre-straining and annealing. *Materials Science and Engineering: A* **647**, 216-221 (2015).

[25] Paper 6 Xu, W. *et al.* A high-specific-strength and corrosion-resistant magnesium alloy. *Nature materials* **14**, 1229-1235 (2015).

[26] Paper 7 Yan, C. *et al.* Evading strength-corrosion tradeoff in Mg alloys via dense ultrafine twins. *Nat Commun* **12**, 4616 (2021).

Reviewer #3:

The corrosion behaviors of Mg-11Y-1Al (wt.%) in chlorine-containing environment were studied by conventional corrosion test and surface analysis. The mechanical and corrosion resistance of the alloy can be greatly improved by adding Al. The research of this manuscript is of high scientific significance and engineering application values. However, the authors do need to address the following concerns before it can be accepted for publication:

Comment 1: The authors use “stainless magnesium alloy” in title to emphasize the corrosion low rate. It is necessary to define “stainless” first. For example, under similar conditions, does the corrosion rate is comparable with “stainless steel” or not? A recently developed Mg-Ca alloy with the corrosion rate of 0.1 mm/year is merely considered to approaching “stainless” (Materials Horizons, 2020). In addition, The authors need to cite and address the following two papers: “A high-specific-strength and corrosion-resistant magnesium alloy (Nature Materials, 2015)” and “Evading strength-corrosion tradeoff in Mg alloys via dense ultrafine twins (Nature Communications, 2021)”.

[Response]: We appreciate that the reviewer pointed out the issue related to the usage of tricky word “stainless”. As a matter of fact, the original idea of proposing a “stainless magnesium alloy” is inspired by the definition of “stainless steel”, because the corrosion rate of the Mg-11Y-1Al was not only low but gradually reached a plateau with the increase of soaking time in the brine solution (**Fig. 3f**). The protective corrosion production film developed on the surface of Mg-11Y-1Al alloy resembles the passive film on the surface of stainless steel. Nevertheless, the absolute value of corrosion rate of Mg-11Y-1Al is still far higher than with stainless steel. So, the usage of “making a high-strength stainless magnesium alloy” in this scenario might cause some confusion to some extent. As a result, we agree to use a more conservative title “approaching a high-strength stainless magnesium alloy” in the revised manuscript. With respect to the two papers (Nature Materials, 2015 and Nature Communications, 2021), we have cited and addressed them along with other papers in the revised introduction part.

Comment 2: Only the yield strength and corrosion rate have been compared to the available statistical data, but the authors draw the conclusions that the combination of the corrosion rate, strength, and elongation of the Mg-11Y-1Al outperforms all the commercial magnesium alloys (Line11-13). The authors need to clarify the reason.

[Response]: We appreciate the reviewer’s suggestion to include comparison of elongation to other Mg alloys. It is well recognized that a high-performance Mg alloy for automotive or aerospace applications usually requires a good combination of high strength, good ductility and excellent corrosion resistance. To date, the strength-corrosion trade-off is the main bottleneck that limits the applications of Mg alloys in a larger scale. This paper aims to solve this long-standing issue, and that is the reason why we mainly compare the strength and corrosion resistance of newly designed Mg-11Y-1Al with other Mg alloys reported in the literature. Regarding the ductility, a minimum 5% elongation is usually required for industrial applications as a rule of thumb. Since the extruded Mg-11Y-1Al alloy presents a moderate tensile elongation of 8%, it also fulfills the basic requirement of ductility. Although the ductility of Mg-11Y-1Al alloy is not outstanding on its own, the combination of strength, corrosion rate and ductility of Mg-11Y-1Al alloy still outperforms all the commercial Mg alloys. For better reflection of the combination of these three properties, **Fig. 1f** has been upgraded to include the data of elongation compared to other Mg alloys, where alloys with elongation lower than 5% are denoted by open symbols while those higher than 5% are denoted by full symbols.

Comment 3: A $Y_2O_3/Y(OH)_3$ fast deposition model is proposed to illustrate the formation of the protective film. However, this model only considers the deposition process of the Mg-11Y and Mg-11Y-1Al, while the dissolve behaviors of Y^{3+} and Al^{3+} , which is equally important, are ignored.

[Response]: We agree with the reviewer that the dissolution behavior of Mg-11Y-1Al alloy is an important aspect in the process of corrosion. When the Al and Y solute atoms

on the sample surface were in contact with the brine solution, they lost electrons and then transformed into Al^{3+} and Y^{3+} ions, respectively, which dissolved into the solution simultaneously. In contrast, the Al and Y atoms in the second phases did not dissolve into the solution because the second phases are cathode phases, as shown in **Fig. S3**. Due to the low solubility of Al^{3+} and Y^{3+} , their concentrations in the brine solution quickly reached saturation, leading to the precipitation of Al and Y hydroxides on the alloy surface and the subsequent formation of protective product film. We have added the discussion of dissolve behaviors of Al and Y atoms in the revised manuscript (See **Line 276-283, Page 14** as follows “When the Al and Y atoms in the matrix of the alloy surface are in contact with the brine solution...”).

Comment 4: The authors tried to interpret the moderate elongation which is rarely mentioned in the main text, while little explanation is given to the high strength which is mentioned in the title, and more likely to be a potential highlight.

[Response]: We appreciate the reviewer’s suggestion to include more interpretation of the moderate elongation and high strength of extruded Mg-11Y-1Al alloy. The moderate elongation is mainly attributed to the following three aspects. Firstly, the reduction of grain size is beneficial on the ductility of magnesium⁹. Secondly, a weak extrusion texture (as shown in **Fig. S4b**) does not deteriorate the ductility of magnesium¹⁰⁻¹². Last but not least, sufficient non-basal slip systems are activated in the extruded Mg-11Y-1Al alloy. The TEM observation shows the dislocation behavior in the T4-1 and T4-EX-1 Mg-11Y-1Al samples, respectively (See **Fig. S7**). The above interpretation has been updated in the main text of the revised manuscript (See **Line 146-151, Page 7**). In addition, the severe stress concentration around Al_2Y particles in solution treated Mg-11Y-1Al alloy (Sample T4-1) has been significantly alleviated by the grain refinement after extrusion (Sample T4-EX-1), thus delaying the fracture of the alloy, as shown in **Fig. R6** from our earlier publication¹³. This part of discussion has been added in the revised supplementary materials (See **Line 116-125, Page 7**).

Fig. R6 (a) (c) Representative volume elements modeling the coarse and fine grain scenarios with Al₂Y particles; (b) and (d) CPFEM simulation results of von Mises stress distribution¹³

The high strength of extruded Mg-11Y-1Al is the result of cooperation of the following strengthening components, including the friction stress of pure Mg (σ_{Mg}), solid-solution strengthening (σ_{ss}), grain-refinement strengthening (σ_{gb}), precipitation strengthening (σ_{ppt}) as well as texture strengthening (σ_{tex}), which can be illustrated via the following equation.

$$\sigma_{0.2} = \sigma_{Mg} + \sigma_{ss} + \sigma_{ppt} + \sigma_{gb} + \sigma_{tex}$$

where $\sigma_{Mg} = 17$ MPa, $\sigma_{ss} = 80$ MPa, $\sigma_{ppt} = 20$ MPa, $\sigma_{gb} = 194$ MPa, $\sigma_{tex} = 33$ MPa. The values of each strengthening component are based on the chemical composition and microstructure features of extruded Mg-11Y-1Al sample and the calculation details has been included in the revised supplementary materials (See Line 57-101, Page 4-6).

Comment 5: It will be great if specific equation can be given to illustrate the effect of Al addition on the improvement of corrosion resistance.

[Response]: Thanks for the suggestion. The addition level of Al is crucial to maintain good corrosion resistance of this series of Mg-Y-Al alloys. In order to show the effect of Al addition on improvement of corrosion resistance in Mg-Y-Al alloy, we have manufactured quite a few additional cast Mg-Y-Al alloys with different Al and Y concentrations and investigated their corrosion performance. The corrosion rates of these Mg-Y-Al alloys were graphically illustrated in the bubble chart of Fig. R7 (S17) below, where the corrosion rate is proportional to the areas of the bubbles. The highest corrosion rates were observed from binary Mg-Y alloys (150 mm/y) and Mg-Al alloys (20 mm/y⁷). In contrast, the corrosion rate decreases dramatically when ternary Mg-Y-

Al alloys are under consideration. The minimum corrosion rate (~ 0.2 mm/y) was found when Al concentration reaches 1 wt. %. Meanwhile, the variation of Y concentration between 4 wt. % and 11 wt. % does not induce noticeable change of the corrosion rate of Mg-Y-1Al alloys. In a nutshell, our current study indicates that the desirable corrosion rate (~ 0.2 mm/y) can be achieved in a wide range of Mg-(4-11)Y-1Al alloys. However, a higher Y addition is preferable due to the better mechanical properties. Accordingly, we have added the following sentences in the revised supplementary materials (See Line 304-321, Page 20-21 as follows “The corrosion rate of Mg-Y-Al alloys also depends on the addition level of Al and Y…”).

Fig. R7 (S17) Effect of different Y and Al additions on corrosion rate of magnesium alloys. The size of the bubbles is proportional to the corrosion rate.

Comment 6: Line 270, “K eV” should be “k eV”.

[Response]: Thanks for pointing out this typo. We have amended it in the revised manuscript.

Comment 7: Line 273, space should be inserted between “50” and “pA”.

[Response]: Thanks. We have amended it as well.

Comment 8: There is no testing parameter for PDP in the experimental section. What is the “standard PDP test”? How to choose the scanning potential range and scanning rate?

[Response]: Thanks for the questions. PDP was conducted from $E_{OCP} - 0.35V$ to E_{OCP}

+ 0.35V with a scan rate of 0.5 mV/s. The selection criterion is that this range and speed reveal the anode and cathode reactions of magnesium well which is very common in magnesium PDP test ¹⁴.

Comment 9: Is the platinum electrode sheet or mesh? What is the specific area?

[Response]: The platinum electrode is a sheet and its specific area is $1 \times 1 \text{ cm}^2$.

Comment 10: What is the size of the electrochemical test sample? Is the potential applied by the EIS direct current or alternating current?

[Response]: The size of the electrochemical test sample was $1 \times 1 \times 0.5 \text{ cm}^3$, which is mounted by the cold resin and the exposed area is $1 \times 1 \text{ cm}^2$. The EIS was measured at OCP in the frequency range from 100 kHz to 10 mHz with an AC amplitude of 10 mV.

Comment 11: In immersion test section, the size of the sample and the immersion time are missing.

[Response]: The size of immersion test sample is $1 \times 1 \times 0.5 \text{ cm}^3$, and the immersion time is 14 days.

Comment 12: The Chinese parentheses in the in-depth analysis diagram should be changed to English.

[Response]: We apologize for this omission and have amended the parentheses in this figure.

Comment 13: The equivalent circuit and fitting line should be given in EIS.

[Response]: The equivalent circuit and corresponding fitting lines are added in **Fig. S13** and **Fig. 3**, respectively.

References:

- 1 Liu, D., Liu, Z. & Wang, E. Effect of rolling reduction on microstructure, texture, mechanical properties and mechanical anisotropy of AZ31 magnesium alloys. *Materials Science and Engineering: A* **612**, 208-213 (2014).
- 2 Li, C.-j. *et al.* Microstructure, texture and mechanical properties of Mg-3.0Zn-0.2Ca alloys fabricated by extrusion at various temperatures. *Journal of Alloys and Compounds* **652**, 122-131 (2015).
- 3 Somekawa, H. & Mukai, T. Hall–Petch relation for deformation twinning in solid solution magnesium alloys. *Materials Science and Engineering: A* **561**, 378-385 (2013).
- 4 Yu, H., Xin, Y., Wang, M. & Liu, Q. Hall-Petch relationship in Mg alloys: A review. *Journal of Materials Science & Technology* **34**, 248-256 (2018).
- 5 Atrous, A. *et al.* Review of Mg alloy corrosion rates. *Journal of Magnesium and Alloys* **8**, 989-998 (2020).
- 6 Esmaily, M. *et al.* Fundamentals and advances in magnesium alloy corrosion. *Progress in Materials Science* **89**, 92-193 (2017).
- 7 Liu, M. *et al.* Calculated phase diagrams and the corrosion of die-cast Mg–Al alloys.

-
- Corrosion Science* **51**, 602-619 (2009).
- 8 Pourbaix M & F, J. A. *Atlas of electrochemical equilibria in aqueous solutions*. (National Association of Corrosion Engineers, 1974).
- 9 Wu, Z., Ahmad, R., Yin, B., Sandlöbes, S. & Curtin, W. A. Mechanistic origin and prediction of enhanced ductility in magnesium alloys. *Science* **359**, 447-452 (2018).
- 10 Sandlöbes, S., Zaeferrer, S., Schestakow, I., Yi, S. & Gonzalez-Martinez, R. On the role of non-basal deformation mechanisms for the ductility of Mg and Mg–Y alloys. *Acta Materialia* **59**, 429-439 (2011).
- 11 Stanford, N., Atwell, D. & Barnett, M. R. The effect of Gd on the recrystallisation, texture and deformation behaviour of magnesium-based alloys. *Acta Materialia* **58**, 6773-6783 (2010).
- 12 Wu, Z., Ahmad, R., Yin, B., Sandlöbes, S. & Curtin, W. A. Mechanistic origin and prediction of enhanced ductility in magnesium alloys. *Science* **359**, 447-452 (2018).
- 13 Zhu, Q. *et al.* Influence of Al₂Y particles on mechanical properties of Mg-11Y-1Al alloy with different grain sizes. *Materials Science and Engineering: A* **831** (2022).
- 14 Xiao, B. *et al.* Achieving Ultrahigh Anodic Efficiency via Single-Phase Design of Mg-Zn Alloy Anode for Mg-Air Batteries. *ACS Appl Mater Interfaces* **13**, 58737-58745 (2021).

REVIEWER COMMENTS

Reviewer #3 (Remarks to the Author):

The manuscript has been improved significantly compared with previous version. The findings that the as-designed magnesium alloy has combined superb corrosion resistance and excellent mechanical properties are of great scientific importance and promising engineering applications. However, the authors still need to address the following issues before the paper be recommended for publication.

1. The mechanism of the excellent corrosion resistance. As illustrated in Fig. R3, the corrosion rate of ternary alloy (Mg-Y-Al) is much lower than binary alloys, such as Mg-Y and Mg-Al. The authors proposed that it was the low solubility of Al^{3+} and Y^{3+} in brine that leads to quickly precipitation of $Al(OH)_3$, and meanwhile the initial deposition of $Al(OH)_3$ may provide nucleation sites for other protective corrosion products like $Y_2O_3/Y(OH)_3$. It will be helpful if the authors can define the “low solubility” of Al^{3+} and Y^{3+} and give specific numbers. Given the experiment verification is too difficult to carry out at current stage, it will be helpful to perform some theoretical calculations to clarify the interaction between Al and Y during dissolve and deposition process.
2. Because the mechanism is not fully confirmed yet, in abstract, it may be more safe to say “our findings are expected to inspire the design of new high performance magnesium alloys” instead of “Such an alloy design strategy can also be applied to other Mg-RE-Al alloy systems ...”
3. Figure 1, please arrange the sub-figures in the sequence they appear in the main text.
4. Fig. 2, please conduct further quantitative in-depth analysis and highlight the important information intended convey to the readers. It is the authors’ responsibility to analyze large amounts of raw data, find novelty and present them in a way that is easy to understand by readers.
5. Fig. 3, same comment as Fig. 2.
6. Fig. 4. A better version is necessary!
7. Line 152: Secondly, a weak extrusion texture (as shown in Fig. S4b) does not deteriorate the ductility of magnesium. This is a very vague description. It may be better to say: “the severe stress concentration around Al_2Y particles in Mg-11Y-1Al alloy is significantly alleviated by the grain refinement caused by the extrusion process, thus delaying the fracture of the alloy”, as has been discussed in Supplementary materials Line 125.
8. In Line 226, since the best comprehensive performance one is the as-extruded Mg-11Y-1Al, why choose the as-cast one for the further corrosion behavior analysis?
9. Comment 8: In Line 94, $250 \text{ MPa}\cdot\text{um}^{-1/2}$ 7,8 -> $252 \text{ MPa}\cdot\text{um}^{-1/2}$. In references 7-8, the value is $252 \text{ MPa}\cdot\text{um}^{-1/2}$, and the $250 \text{ MPa}\cdot\text{um}^{-1/2}$ is calculated before.
10. In Fig.S13 and Table.S1, The title for Table S1 is “The parameter we adopted in the simulation of equivalent circuit diagram” and R_f is one of these parameters which seem to be an input variable. However, R_f seems to be used as an output variable during discussion. In addition, the number of significant digits in Table S1 should be corrected
11. Line 368, the area should be $1\times 1\text{cm}^2$.
12. In the PDP test section, the test range is EOC_P - 0.35V to EOC_P + 0.35V, but it seems obviously not the same according to the test results in Fig. 3
13. The chemical composition of AZ91D in salt spray test should be given in Table S2.

14. Please check the the unit of R_f in line 222 of the supplementary material.

Reviewer #4 (Remarks to the Author):

The manuscript describes work on what the authors refer to as “stainless” Mg alloys. Based on the corrosion rates observed (ca. 0.2 mm/yr), I think the adjective is a bit of hyperbole. Certainly the corrosion resistance is improved, but in the ferrous case, the decrease in corrosion rate is to a far greater extent. The name will catch people’s attention, but it is a bit misleading.

The authors’ responses to the reviewers’ comments about corrosion data are well thought out for the most part. Although I don’t recommend additional experiments, the authors could consider a few points. These are optional, in my opinion.

1. The authors dance around why the LPSO and Al₂Y phases are both very noble to the matrix, but have very different impacts on the corrosion of the matrix. If the presence of Al in the matrix is the key, then adding Al ions to the bulk solution should render the same effect on the Al-11Y alloy.
2. In Figure S11, the authors state that the outer film is MgO, but there does not appear to be much Mg in the outer portion of the film, and no diffraction results are presented as they are for the Y₂O₃. The XPS of Figure 3 is supportive, but why wasn’t a spectrum shown of the Mg peak in the outer layer?
3. The authors state that in Figure S14a “the hydrogen evolution volume of the Mg-11Y-1Al alloy in different states all increases slowly with immersion time. However, the difference of the hydrogen volume between three specimens is marginal, indicating that the corrosion resistance of Mg-11Y-1Al alloy shows low sensitivity to microstructure due to formation of the protective film layer on the alloy surface.” However, the results are well outside the presented error bars. Wouldn’t that, by definition, mean that the differences were statistically significant?
4. I think the authors miss a key result from Figure 3d. It is clear from the figure that the addition of the Al causes a large movement of the corrosion potential in the negative direction and a simultaneous decrease in the corrosion rate. Mixed Potential Theory states that the only way in which that can happen is if there is suppression of the cathodic reaction dominating. The authors seem focused on the effect of Al on the dissolution rate, but Figure 3d indicates that its strongest effect is on suppression of the hydrogen evolution reaction to such an extent that the potential falls below the pitting potential of ~ -1.55 V(SCE). There is literature that has shown that Al³⁺ ions suppress hydrogen evolution (C. Liu, P. Khullar, R. G. Kelly, “Acceleration of the Cathodic Kinetics on Aluminum Alloys by Aluminum Ions,” J. Electrochem. Soc. 166(6), C153-C161 (2019). <https://doi.org/10.1149/2.0571906jes>) which may be what is happening here.

In summary, the authors sufficiently address the reviewers’ comments that publication is recommended after they authors consider the optional revisions recommended above.

Point by point response to reviewers' comments

Dear reviewers,

We sincerely appreciate the invaluable and constructive comments from you. The manuscript has been revised thoroughly to incorporate all the comments. In addition, some figures have been adjusted in the manuscript and supplementary materials to address the suggestions. **The amendments were highlighted in red in the revised manuscript and supplementary materials** (the order of figures in the manuscript and supplementary materials were also adjusted to reflect the relevant changes in the main text), and a point-by-point response is enclosed below. We hope the revised version is now suitable for publication in *Nature communications*.

Yours Sincerely
Yangxin Li

Reviewer #3 (Remarks to the Author):

The manuscript has been improved significantly compared with previous version. The findings that the as-designed magnesium alloy has combined superb corrosion resistance and excellent mechanical properties are of great scientific importance and promising engineering applications. However, the authors still need to address the following issues before the paper be recommended for publication.

Comment 1: The mechanism of the excellent corrosion resistance. As illustrated in Fig. R3, the corrosion rate of ternary alloy (Mg-Y-Al) is much lower than binary alloys, such as Mg-Y and Mg-Al. The authors proposed that it was the low solubility of Al^{3+} and Y^{3+} in brine that leads to quickly precipitation of $\text{Al}(\text{OH})_3$, and meanwhile the initial deposition of $\text{Al}(\text{OH})_3$ may provide nucleation sites for other protective corrosion products like $\text{Y}_2\text{O}_3/\text{Y}(\text{OH})_3$. It will be helpful if the authors can define the “low solubility” of Al^{3+} and Y^{3+} and give specific numbers. Given the experiment verification is too difficult to carry out at current stage, it will be helpful to perform some theoretical calculations to clarify the interaction between Al and Y during dissolve and deposition process.

[Response]: We appreciate this comment and agree that it would be more convincing to define the “low solubility” of Al^{3+} and Y^{3+} with specific numbers. We are also very grateful for your understanding of the experiment difficulty at current stage. According to the solubility product constants (K_{sp}) of various ions in solution¹, the saturated solubility of ions in brine solution at 25°C with a certain pH value can be obtained according the following formula: $K_{\text{sp}} = c(\text{A}^{m+})^n \times c(\text{B}^{n-})^m$. When a magnesium alloy is soaked in brine, the pH value of its surface is about 10.33^{2,3} and the approximate saturated solubility of Al^{3+} and Y^{3+} ions are about 1.33×10^{-22} mol/L and 8.19×10^{-12} mol/L, respectively. Such a low solubility of Y^{3+} and Al^{3+} ions in brine indicates that the precipitation of $\text{Y}(\text{OH})_3$ and $\text{Al}(\text{OH})_3$ is preferential if the Y and Al elements are dissolved from the surface. Given that the Y content in the surface film of ternary Mg-11Y-1Al alloy is much higher than that of binary Mg-11Y alloy (**Fig. S11**), and the corrosion products on the surface of Mg-11Y-

1Al alloy developed much more quickly than that on the surface of Mg-11Y alloy (**Fig. S16**), it is reasonable to conclude that the deposition process of $Y(OH)_3$ is ineffective without the help of $Al(OH)_3$. Then, the initial deposition of $Al(OH)_3$ is probable to act as nucleation sites for $Y(OH)_3$, which leads to a much higher content of Y in the dense surface film of Mg-11Y-1Al alloy. Accordingly, we have given specific numbers to define the “low solubility” of Al^{3+} and Y^{3+} in the revised manuscript (See Line 286-290, Page 14 in the revised manuscript as follows “When a magnesium alloy is soaked in brine, the pH value of its surface is about 10.33 and the approximate saturated solubilities of Al^{3+} and Y^{3+} ions are about 1.33×10^{-22} mol/L and 8.19×10^{-12} mol/L, respectively.”).

Fig. R1 Solubility curves of Mg, Y and Al in brine solution vs $pH^{2,3}$.

Comment 2: Because the mechanism is not fully confirmed yet, in abstract, it may be more safe to say “our findings are expected to inspire the design of new high performance magnesium alloys” instead of “Such an alloy design strategy can also be applied to other Mg-RE-Al alloy systems ...”

[Response]: Thanks for the suggestion and we have amended the description in the abstract accordingly (See Line 17-18, Page 1 in the revised manuscript as follows “Our findings are expected to inspire the design of new high performance magnesium alloys.”).

Comment 3: Figure 1, please arrange the sub-figures in the sequence they appear in the main text.

[Response]: Thanks for the advice. **Fig. 1** has been updated, of which the clockwise arrangement of the sub-figures is consistent with the sequence they appear in the main text. In addition, we have replaced the OM images with SEM-BSE images.

Comment 4: Fig. 2, please conduct further quantitative in-depth analysis and highlight the important information intended convey to the readers. It is the authors’ responsibility to analyze

large amounts of raw data, find novelty and present them in a way that is easy to understand by readers.

[Response]: Thanks for this constructive comment and we agree that Fig. 2 is a bit overcrowded. In the revised manuscript, **Fig. 2** has been tidied up and only the EDS maps of corrosion product films are reserved while the EPMA results are moved to **Fig. S11** in the supplementary materials. We have also rearranged the EDS maps to make it easier for readers to find out the difference in the corrosion product film between Mg-11Y and Mg-11Y-1Al alloys.

Comment 5: Fig. 3, same comment as Fig. 2.

[Response]: We have divided **Fig. 3** into two figures, i.e., new **Fig. 3** (XPS analysis) and new **Fig. 4** (electrochemical behavior) in the revised manuscript. The key information has been highlighted in detail (See Line 178-184, Page 9 and Line 242-245, Page 12 in the revised manuscript as follows “This thin and uniform corrosion product also contains Mg, Y and O elements while the Al element is hard to be detected with EDS in such a scale.” and “The mixed potential theory indicates that such a significant reduction is attributed to the strong suppression of the cathodic hydrogen evolution reaction, which is also confirmed by the hydrogen evolution volume in **Fig. 1a**.”).

Comment 6: Fig. 4. A better version is necessary!

[Response]: We have reconstructed the schematic diagram in Fig. 4 (new Fig. 5 in the revised manuscript), which provides a clearer picture showing the difference of corrosion mechanism between Mg-11Y and Mg-11Y-1Al alloys after a short and long period of immersion test.

Comment 7: Line 152: Secondly, a weak extrusion texture (as shown in Fig. S4b) does not deteriorate the ductility of magnesium. This is a very vague description. It may be better to say: “the severe stress concentration around Al₂Y particles in Mg-11Y-1Al alloy is significantly alleviated by the grain refinement caused by the extrusion process, thus delaying the fracture of the alloy”, as has been discussed in Supplementary materials Line 125.

[Response]: Thanks for pointing out this specific issue. We have revised the description accordingly (See Line 145-147, Page 7 in the revised manuscript as follows “Secondly, the severe stress concentration around Al₂Y particles in Mg-11Y-1Al alloy is significantly alleviated by the grain refinement caused by the extrusion process, thus delaying the fracture of the alloy.”).

Comment 8: In Line 226, since the best comprehensive performance one is the as-extruded Mg-11Y-1Al, why choose the as-cast one for the further corrosion behavior analysis?

[Response]: This is a very good question. The reasons are two-fold. Firstly, the corrosion rate of Mg-11Y-1Al alloy shows low sensitivity to its microstructure. The as-cast one still has the same level of corrosion resistance as the extruded one. Secondly, the microstructure after extrusion is far more complicated than as-cast one because of the presence of precipitates and forest dislocations (**Fig. S6**), which will increase the difficulty of probing the mechanism of its corrosion behavior.

Comment 9: In Line 94, $250 \text{ MPa}\cdot\text{um}^{-1/2}$ ^{7,8} -> $250 \text{ MPa}\cdot\text{um}^{-1/2}$. In references ⁷⁻⁸, the value is $252 \text{ MPa}\cdot\text{um}^{-1/2}$, and the $250 \text{ MPa}\cdot\text{um}^{-1/2}$ is calculated before.

[Response]: Thanks for pointing out this issue. The k value is estimated to be close to $252 \text{ MPa}\cdot\text{um}^{-1/2}$ (see section 3 in the supplementary materials), which is one of the commonly used k

value of Mg-Y based alloys in literature. We apologize for rounding off the value from 252 to 250 in the previous text. We have amended the description to make it consistent in the revised manuscript and the supplementary materials. At the same time, some related values are recalculated and updated in the revised supplementary materials, as highlighted in red.

Comment 10: In Fig.S13 and Table.S1, The title for Table S1 is “The parameter we adopted in the simulation of equivalent circuit diagram” and Rf is one of these parameters which seem to be an input variable. However, Rf seems to be used as an output variable during discussion. In addition, the number of significant digits in Table S1 should be corrected

[Response]: We appreciate the reviewer’s suggestion. We have made corresponding corrections in the revised version. Herein, Rf is an input variable for numerical simulation, but its value will be updated after each round of iteration. So, the result shown in Table S1 is the optimum Rf value after simulation. That is why it looks like an output variable in our discussion. In addition, we have reserved three significant digits invariantly in Table S1 when the numerical results are expressed in scientific notation.

Comment 11: Line 368, the area should be $1 \times 1 \text{ cm}^2$.

[Response]: We apologize for this omission and have amended it in the revised manuscript (See Line 368 and 370, Page 18-19 in the revised manuscript).

Comment 12: In the PDP test section, the test range is EOCV - 0.35V to EOCV + 0.35V, but it seems obviously not the same according to the test results in Fig. 3

[Response]: We apologize for this mistake and have amended the corresponding parameters to be EOCV - 0.4 V to EOCV + 0.4 V. (See Line 371, Page 19 in the revised manuscript).

Comment 13: The chemical composition of AZ91D in salt spray test should be given in Table S2.

[Response]: We have added the ICP data of selected AZ91D alloy in the revised Table S2. (See Line 327, Page 21 in the revised supplementary materials).

Comment 14: Please check the the unit of Rf in line 222 of the supplementary material.

[Response]: We apologize for this omission and have amended it in the revised supplementary materials. (See Line 226, Page 15 in the revised supplementary materials)

Reviewer #4 (Remarks to the Author):

The manuscript describes work on what the authors refer to as “stainless” Mg alloys. Based on the corrosion rates observed (ca. 0.2 mm/yr), I think the adjective is a bit of hyperbole. Certainly the corrosion resistance is improved, but in the ferrous case, the decrease in corrosion rate is to a far greater extent. The name will catch people’s attention, but it is a bit misleading.

[Response]: We appreciate that the reviewer pointed out the issue related to the usage of tricky word “stainless”. As a matter of fact, the original idea of proposing a “stainless magnesium alloy” is

inspired by the corrosion behavior of “stainless steel”, because the corrosion rate of the Mg-11Y-1Al was not only low but gradually reached a plateau with the increase of soaking time in the brine solution (**Figs. 4c and S15**). In this regard, the protective corrosion product film developed on the surface of Mg-11Y-1Al alloy, to some extent, resembles the passive film on the surface of stainless steel. Nevertheless, we admit that the absolute value of corrosion rate of Mg-11Y-1Al is still far higher than that of stainless steel. So, instead of using “making a high-strength stainless magnesium alloy”, we have taken a step backward and chosen a more conservative title “approaching a high-strength stainless magnesium alloy”. A similar description can be found in another recent publication [19], Deng, M. *et al.* Approaching “stainless magnesium” by Ca micro-alloying. *Materials Horizons* **8**, 589-596 (2021)), where a similarly low corrosion rate was reported and the term “stainless magnesium” was only used to demonstrate its superior corrosion resistance over the other Mg alloys reported in the literature.

The authors’ responses to the reviewers’ comments about corrosion data are well thought out for the most part. Although I don’t recommend additional experiments, the authors could consider a few points. These are optional, in my opinion.

Comment 1: The authors dance around why the LPSO and Al₂Y phases are both very noble to the matrix, but have very different impacts on the corrosion of the matrix. If the presence of Al in the matrix is the key, then adding Al ions to the bulk solution should render the same effect on the Al-11Y alloy.

[Response]: Thanks for this great suggestion. In fact, we have briefly studied the effect of adding Al³⁺ to the solution on the corrosion rate of Mg-11Y alloy. After adding a small amount of Al(OH)₃ into 3.5 wt. % NaCl solution to reach a pH 10.33 condition, the hydrogen evolution rate of Mg-11Y alloy was tested and compared with that without Al³⁺ addition at the same pH condition. As shown in **Fig. R2**, the hydrogen evolution rate reduced by 50% after being soaked for 5 hours when Al³⁺ are present. This result indicates that the addition of Al³⁺ into a brine solution does render the same effect on the Mg-11Y alloy, but this approach is far less effective than micro-alloying Al into the Mg matrix. The potential reasons of its low effectiveness are two-fold. Firstly, the added Al³⁺ are continuously consumed after deposition at the surface of Mg-11Y alloy sample while Al³⁺ can be dynamically replenished from the interior matrix of Mg-11Y-1Al alloy sample; Secondly, the microstructure of Mg-11Y alloy is significantly changed after micro-alloying Al, which induces substantial grain refinement. The galvanic corrosion of Mg-11Y alloy is more intensive than that of Mg-11Y-1Al alloy.

Fig. R2 Effect of Al(OH)₃ addition into 3.5 wt. % NaCl solution on hydrogen evolution rate of as-cast Mg-11Y (W11) alloy in the early corrosion stage.

Comment 2: In Figure S11, the authors state that the outer film is MgO, but there does not appear to be much Mg in the outer portion of the film, and no diffraction results are presented as they are for the Y₂O₃. The XPS of Figure 3 is supportive, but why wasn't a spectrum shown of the Mg peak in the outer layer?

[Response]: Thanks for pointing out this omission. Generally speaking, the outer layer of corrosion product film in most Mg alloys immersing in brine solution is initially formed as Mg(OH)₂, which is usually transformed into MgO after dehydration⁴. When the MgO/Mg(OH)₂ layer is well-established, the Mg atoms on the surface are fully expended. Therefore, the XPS results of Fig. 3 only detect the MgO peak in the outer layer. For Figure S11, we have also added a selected area electron diffraction (SEAD) pattern of the outer layer in **Fig. R3a (new Figure S12a)**, confirming that the outer layer is mainly comprised of MgO.

Fig. R3 Characterization of corrosion product film of the as-cast Mg-11Y-1Al alloy after soaking in 3.5 wt. % NaCl solution for 1 hour: (a) HAADF-STEM image with SEAD patterns from inner layer (Y_2O_3) and outer layer (MgO); (b-d) EDS Mapping.

Comment 3: The authors state that in Figure S14a “the hydrogen evolution volume of the Mg-11Y-1Al alloy in different states all increases slowly with immersion time. However, the difference of the hydrogen volume between three specimens is marginal, indicating that the corrosion resistance of Mg-11Y-1Al alloy shows low sensitivity to microstructure due to formation of the protective film layer on the alloy surface.” However, the results are well outside the presented error bars. Wouldn’t that, by definition, mean that the differences were statistically significant?

[Response]: We agree with the reviewer that the results shown in Figure S15a does indicate there is statistically significant difference in the corrosion rate between various states of Mg-11Y-1Al alloy. Nevertheless, such a difference reflected in Figure S14a is much lower than that of other commonly used Mg-RE or Mg-Al alloys, as shown in Fig. R4. For the other Mg alloys listed in Fig. R4, the corrosion rate between different states usually differs by one order of magnitude, which is much greater than that of Mg-11Y-1Al alloy. In order to avoid any conceivable misunderstanding, we have changed the sentence “the corrosion resistance of Mg-11Y-1Al alloy shows low sensitivity

to microstructure” to “the corrosion resistance of WA111 alloy shows lower sensitivity to microstructure compared to other commonly used Mg-Al or Mg-RE alloys”^{5, 6, 7} (See line 245-246, page 16 in the revised supplementary materials)

Fig. R4 Corrosion rates of several typical Mg alloys with different microstructure^{5, 6, 7}.

Comment 4: I think the authors miss a key result from Figure 3d. It is clear from the figure that the addition of the Al causes a large movement of the corrosion potential in the negative direction and a simultaneous decrease in the corrosion rate. Mixed Potential Theory states that the only way in which that can happen is if there is suppression of the cathodic reaction dominating. The authors seem focused on the effect of Al on the dissolution rate, but Figure 3d indicates that its strongest effect is on suppression of the hydrogen evolution reaction to such an extent that the potential falls below the pitting potential of ~ -1.55 V(SCE). There is literature that has shown that Al^{3+} ions suppress hydrogen evolution (C. Liu, P. Khullar, R. G. Kelly, “Acceleration of the Cathodic Kinetics on Aluminum Alloys by Aluminum Ions,” J. Electrochem. Soc. 166(6), C153-C161 (2019), <https://doi.org/10.1149/2.0571906jes>) which may be what is happening here.

[Response]: Many thanks to the reviewer’s constructive comment. This enlightening literature does provide a new angle for us to better understand the corrosion behavior of such a novel Mg-11Y-1Al alloy. Accordingly, we have added the description in the revised manuscript (See Line 242-244, Page 12 as follows “The mixed potential theory indicates that such a significant reduction is attributed to the strong suppression of the cathodic hydrogen evolution reaction, which is also confirmed by the hydrogen evolution volume in Fig. 1a.”).

References:

1. Dean JA. *Lange's Handbook of Chemistry (15th Edition)*. Lange's Handbook of Chemistry (15th Edition) (1999).

2. Lindström R, Johansson L-G, Thompson GE, Skeldon P, Svensson J-E. Corrosion of magnesium in humid air. *Corrosion Science* **46**, 1141-1158 (2004).
3. Shahabi-Navid M, *et al.* NaCl-Induced Atmospheric Corrosion of the MgAl Alloy AM50-The Influence of CO₂. *Journal of The Electrochemical Society* **161**, C277-C287 (2014).
4. Song G-L, Unocic KA. The anodic surface film and hydrogen evolution on Mg. *Corrosion Science* **98**, 758-765 (2015).
5. Xiao B, Song G-L, Zheng D, Cao F. A corrosion resistant die-cast Mg-9Al-1Zn anode with superior discharge performance for Mg-air battery. *Materials & Design* **194**, (2020).
6. Cao F-f, Deng K-k, Nie K-b, Kang J-w, Niu H-y. Microstructure and corrosion properties of Mg-4Zn-2Gd-0.5Ca alloy influenced by multidirectional forging. *Journal of Alloys and Compounds* **770**, 1208-1220 (2019).
7. Shi Z, Cao F, Song G-L, Liu M, Atrens A. Corrosion behaviour in salt spray and in 3.5% NaCl solution saturated with Mg(OH)₂ of as-cast and solution heat-treated binary Mg-RE alloys: RE=Ce, La, Nd, Y, Gd. *Corrosion Science* **76**, 98-118 (2013).

REVIEWERS' COMMENTS

Reviewer #3 (Remarks to the Author):

The authors has addressed the reviewers' concerns and publication is recommended!

Reviewer #4 (Remarks to the Author):

The authors have appropriately addressed all of the issues raised in the review. I recommend publication of the revised version.